# Observation of the nonanalytic behavior of optical phonons in monolayer hexagonal boron nitride

Jiade Li[1,2], Li Wang[1], Yani Wang[3,4], Zhiyu Tao[1,2], Weiliang Zhong[1,2], Zhibin Su[1,2], Siwei Xue[1], Guangyao Miao ®[1], Weihua Wang ®[1], Hailin Peng ®[3,4], Jiandong Guo ®[1,2] ✉ & Xuetao Zhu ®[1,2] ✉

Phonon splitting of the longitudinal and transverse optical modes (LO-TO splitting), a ubiquitous phenomenon in three-dimensional polar materials, will break down in two-dimensional (2D) polar systems. Theoretical predictions propose that the LO phonon in 2D polar monolayers becomes degenerate with the TO phonon, displaying a distinctive "V-shaped" nonanalytic behavior near the center of the Brillouin zone. However, the full experimental verification of these nonanalytic behaviors has been lacking. Here, using monolayer hexagonal boron nitride (h-BN) as a prototypical example, we report the comprehensive and direct experimental verification of the nonanalytic behavior of LO phonons by inelastic electron scattering spectroscopy. Interestingly, the slope of the LO phonon in our measurements is lower than the theoretically predicted value for a freestanding monolayer due to the screening of the Cu foil substrate. This enables the phonon polaritons in monolayer h-BN/Cu foil to exhibit ultra-slow group velocity ($\sim 5 \times 10^{-6}$ $c$, $c$ is the speed of light) and ultra-high confinement ($\sim$ 4000 times smaller wavelength than that of light). These exotic behaviors of the optical phonons in h-BN presents promising prospects for future optoelectronic applications.

Optical phonons, the out-of-phase collective vibrations of lattice, play essential roles in the optical[1,2], electronic[3,4], and thermal[5,6] properties of crystalline materials. In particular, the behaviors of longitudinal optical (LO) phonons are distinctive due to the polarity of the materials. In non-polar materials, whether in three-dimensional (3D) or two-dimensional (2D) systems, the lattice symmetry guarantees that the LO and the transverse optical (TO) phonons are degenerate with zero dispersion slopes at the center of the Brillouin zone (CBZ) (Fig. 1a). However, in polar materials, the lattice vibrations of the LO phonons generate extra long-range electric fields, which in turn exert long-range Coulomb forces on the polar lattices. The long-range Coulomb interaction significantly changes the behaviors of the LO

phonons. In the 3D scenario, the long-range Coulomb interaction raises the frequency of the LO phonon near the CBZ, leading to an energetic split with the TO phonon, known as the LO-TO splitting (Fig. 1b). The LO-TO splitting ubiquitously exists in 3D polar materials, with experimental observations reported in GaP[7], SiC[8], BN[9], etc. In the 2D scenario, however, the LO phonon does not split with the TO phonon anymore. Over the last two decades, various theoretical models predict that, in 2D polar monolayers, the LO phonon is degenerate with the TO phonon, exhibiting a "V-shaped" nonanalytic behavior near the CBZ (Fig. 1c)[10–14].

On the other hand, it has been theoretically demonstrated that the nonanalytic behavior of LO phonons in 2D polar monolayers

[1]Beijing National Laboratory for Condensed Matter Physics and Institute of Physics, Chinese Academy of Sciences, 100190 Beijing, China. [2]School of Physical Sciences, University of Chinese Academy of Sciences, 100049 Beijing, China. [3]Center for Nanochemistry, Beijing Science and Engineering Center for Nanocarbons, Beijing National Laboratory for Molecular Sciences, College of Chemistry and Molecular Engineering, Peking University, 100871 Beijing, China. [4]Beijing Graphene Institute (BGI), 100095 Beijing, China. ✉ e-mail: jdguo@iphy.ac.cn; xtzhu@iphy.ac.cn

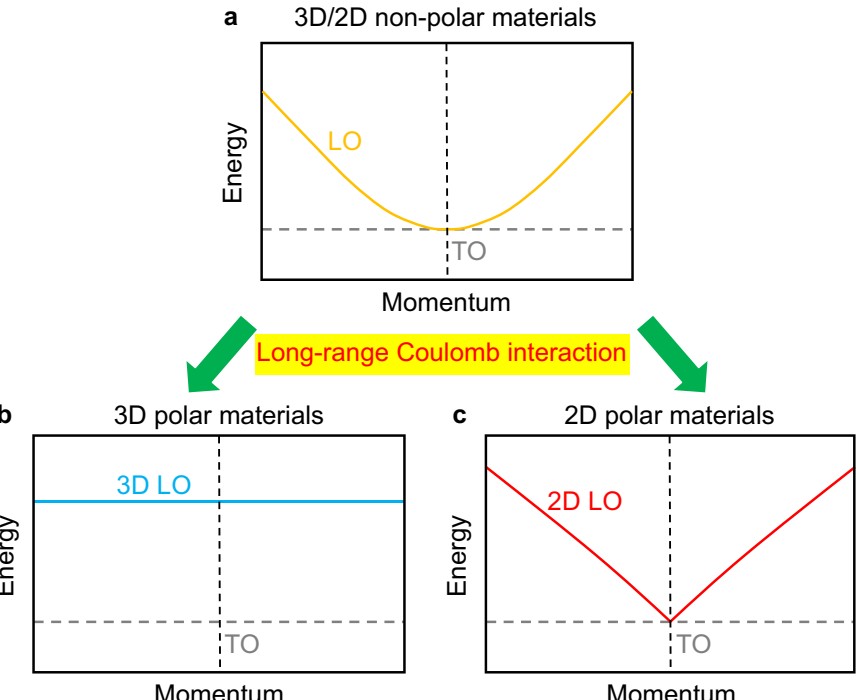

**Fig. 1 | Schematic of the behaviors of the LO phonons near the CBZ. a**–**c** The behaviors of the LO phonons in 3D/2D non-polar, 3D polar, and 2D polar materials, respectively. The dashed black lines represent the CBZ.

makes them equivalent to phonon polaritons (PhPs)[15]. The 2D PhPs are predicted to have excellent characteristics such as ultra-slow group velocity and ultra-high wavelength confinement of light[15,16]. However, existing measurements of LO phonons or PhPs in 2D polar monolayers fall short of capturing the complete dispersion behavior from the CBZ to a large momentum range. Consequently, they are unable to provide a comprehensive understanding of the inclusive properties of PhPs. Raman scattering has revealed the degeneracy of LO and TO phonons at the CBZ[17], but it fails to detect the linear dispersion of LO phonons due to the lack of momentum resolution. Using scattering-type scanning near-field optical microscopy (s-SNOM)[18], the linear dispersion of PhP has been measured[19], but within a confined momentum range (<0.005 Å$^{-1}$). Furthermore, as s-SNOM cannot access the TO mode and the CBZ, it cannot confirm the degeneracy of LO and TO phonons. Another powerful technique for PhP measurements is the electron energy loss spectroscopy incorporated in a scanning transmission electron microscope (STEM-EELS)[20], yet it is also restricted to a small momentum range near the CBZ, and measuring real monolayers remains a significant challenge. Hence, there is an urgent need for direct experimental investigation into the nonanalytic behavior of LO phonons and the properties of PhPs spanning from the CBZ to a large momentum range in strict 2D polar monolayers. Such research is crucial for both fundamental physics and potential applications.

Monolayer hexagonal boron nitride (h-BN), a prototypical 2D polar material, is an ideal candidate for the experimental study of the behaviors of the LO phonons and PhPs in 2D monolayers. Especially, a recent theory predicts[16] that the 2D PhP of a freestanding monolayer h-BN exhibits an ultra-slow group velocity (~10$^{-5}$ $c$, $c$ is the speed of light) and ultra-high confinement (~1000 times smaller wavelength than that of light in free space), indicating great potential for optoelectronic applications. High-resolution electron energy loss spectroscopy (HREELS), highly sensitive to excitations on surfaces[21], is an ideal method for measuring phonon dispersions of 2D monolayers. Especially, the development of 2D-HREELS[22,23] that can perform 2D measurements of energy and momentum simultaneously

with high resolutions [see Methods and Supplementary Fig. S1 in the Supplementary Information (SI) for details], brings new opportunities to observe the behaviors of the LO phonons in 2D polar materials. Here, using monolayer h-BN as a prototypical example, and employing the state-of-the-art 2D-HREELS technique, we systematically observe the complete dispersion behaviors of all unique phonon modes from the CBZ to the Brillouin zone boundary, and investigate the properties of the 2D PhPs in polar monolayer systems.

## Results

### Sample characterization and phonon spectra

Monolayer h-BN was synthesized on a substrate of polycrystalline copper foil (h-BN/Cu foil) and Cu(111) surface of a copper single crystal (h-BN/Cu crystal) (Methods). As shown in Fig. 2a, b, the high crystallographic quality of h-BN was confirmed by the atom-resolved scanning tunneling microscope (STM) image and sharp low-energy electron diffraction (LEED) pattern. For the h-BN/Cu foil sample, it is important to note that the surface roughness of the Cu foil substrate introduces nonuniform background features in STM and LEED images. Due to the constraints associated with the commercial cold rolling process and the flexibility of thin Cu foil, the surface roughness of the substrate is unavoidable. This aspect distinctly affects the phonon properties of h-BN, a point that will be further elucidated later in the subsequent discussions.

The 2D-HREELS measurements were performed at room temperature using an incident electron beam with an energy of 110 eV and an incident angle of 60°. Figure 2c shows the 2D energy and momentum mappings for the phonon spectra of the monolayer h-BN on Cu foil along the Γ-M and Γ-K directions. Unlike the HREELS spectra measured for graphene[24–26] where the non-polar nature makes the phonon intensity distribution relatively uniform in momentum space, the signal intensity of the phonon dispersions of h-BN in the impact scattering region (away from the Γ point, Fig. 2f, g) is suppressed by the strong dipole scattering (around the Γ point, Fig. 2e) due to the polarity of h-BN. Nevertheless, we managed to capture all the phonon modes of

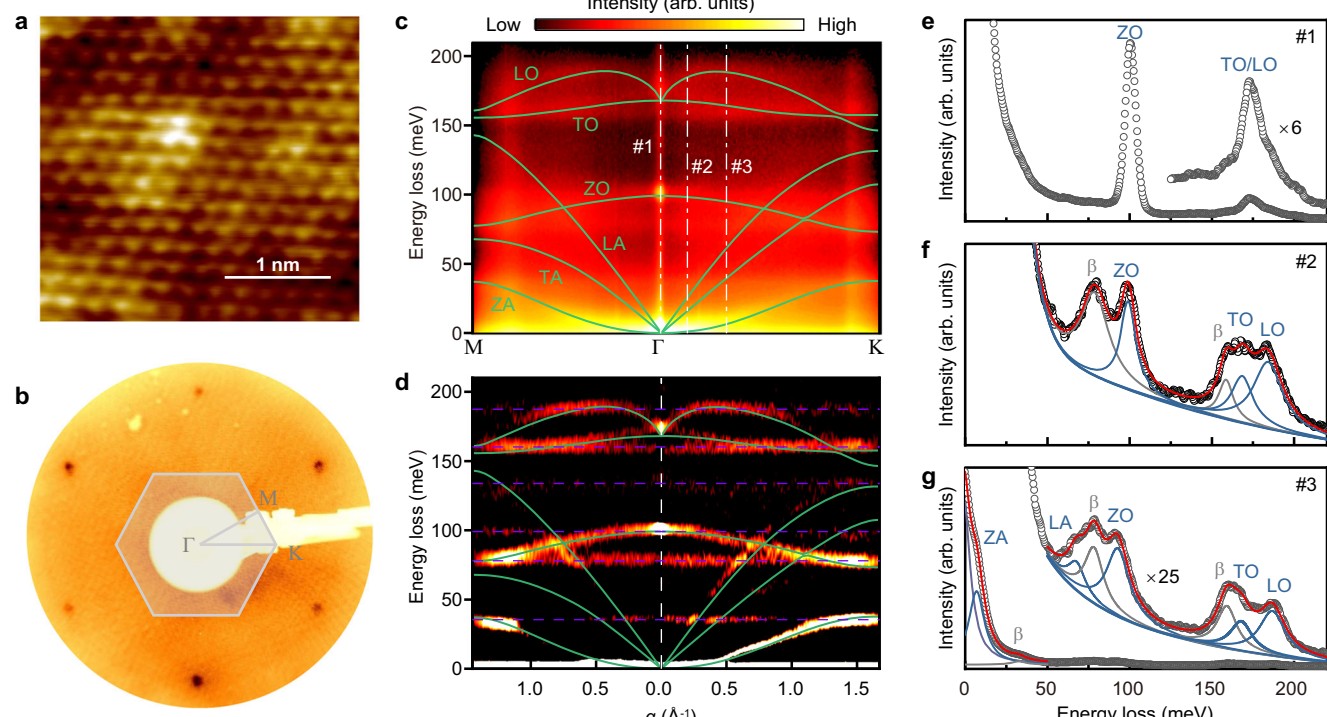

**Fig. 2 | Crystallographic quality and phonon spectra of monolayer h-BN/Cu foil.** **a** Atomic-resolution STM image (scanned at 0.9 V and 0.3 nA). **b** LEED pattern, obtained at room temperature with an incident beam energy of 140 eV. **c** 2D energy-momentum mappings of 2D-HREELS along the Γ-M and Γ-K directions. **d** The second derivative results correspond to **b**. The purple dashed lines mark the replica signals of the phonon. **e**–**g** EDCs corresponding to the #1 (specular direction), #2 (momentum 0.2 Å⁻¹), and #3 (momentum 0.5 Å⁻¹) dashed dot lines in **b**, respectively. The green curves in **c** and **d** are calculated phonon dispersions by considering the 2D implementation of the nonanalytical-term correction[27,28].

monolayer h-BN [see Fig. 2d, the corresponding second derivative results of Fig. 2c (Methods)]. To assign the phonon modes measured by 2D-HREELS, the calculated phonon dispersions, with the 2D implementation of the density-functional perturbation theory (DFPT) developed by Sohier et al.[12,27,28], are also superimposed in Fig. 2c, d (green curves, see Methods for details). The corresponding LO, TO, out-of-plane optical (ZO), transverse acoustic (TA), longitudinal acoustic (LA), and out-of-plane acoustic (ZA) modes are labeled in the calculated phonon dispersions. It can be seen from Fig. 2d that the experimental results are in good agreement with the calculations, and all six phonon modes are identified along the Γ-K direction (although the signal intensity of the TA mode is very weak). Especially, our calculated LO phonon of monolayer h-BN exhibits a "V-shaped" nonanalytical behavior near the Γ point, which originates from long-range Coulomb interactions in 2D polar materials, reproducing the results of previous studies[12,29].

We also notice that there are some dispersionless scattering signals in Fig. 2d (marked by the purple dashed line), which show obvious loss peaks in the energy distribution curves (EDCs) (marked as β peaks in Fig. 2f, g). After careful analyses (discussed in detail in the SI), these signals are proven to be the phononic replicas caused by the surface roughness of the Cu foil substrate rather than the true phonon dispersions of monolayer h-BN. Fortunately, the phononic replicas are uniformly distributed in momentum space like diffuse scattering[21], and thus the dispersions of the monolayer h-BN are unaffected and still well discernable.

## Breakdown of the LO-TO splitting

To illustrate the measured dispersion of the LO phonon of monolayer h-BN, in Fig. 3a we show a zoom-in view of Fig. 2d around the LO phonon. The dispersion of the LO phonon is unambiguously demonstrated and shows a distinct "V-shaped" nonanalytic behavior near the

Γ point. First, the LO and TO phonons are undoubtedly degenerate at the Γ point (see also in Fig. 2e for the EDC at the Γ point). These two branches gradually separate from each other as the momentum increases away from the Γ point. Second, the dispersion of the LO phonon shows a finite positive slope around the Γ point. Our observation, with experimental visualization of the phonon dispersions, directly and comprehensively verifies the physical picture shown in Fig. 1c.

The nonanalytic behavior of the LO phonon originates from the long-range Coulomb interaction caused by the polar lattice vibrations of monolayer h-BN. The modulation of phonon dispersions by long-range Coulomb interactions is greatly affected by dimensionality. Compared with the calculated LO phonon dispersion results of monolayer h-BN under traditional 3D boundary periodic conditions (Supplementary Fig. S5 in SI), our measurement results give a comprehensive experimental verification of the 2D implementation method reported by Sohier et al.[12,27,28].

## Screening effect from Cu foil

The behaviors of the LO phonons in the polar monolayers can be significantly influenced by electronic screening. The Cu foil substrate, as revealed by our investigation, exhibits unique characteristics. In Fig. 3b, the phonon dispersion extracted from the 2D-HREELS measurement is presented. Interestingly, our results indicate that the group velocity of the LO phonon at the Γ point is about $5 \times 10^{-6} c$, significantly lower than the theoretically predicted value of $1.2 \times 10^{-4} c$ for a freestanding monolayer h-BN[15]. This observation suggests that the Cu foil substrate partially screens the polarization electric field of h-BN, leading to a partial suppression of the slope of LO phonons. However, it is well-known that metal substrates typically induce strong screening effects, potent enough to completely suppress the linear dispersion of LO phonons around the Γ point. To comprehend the distinctive

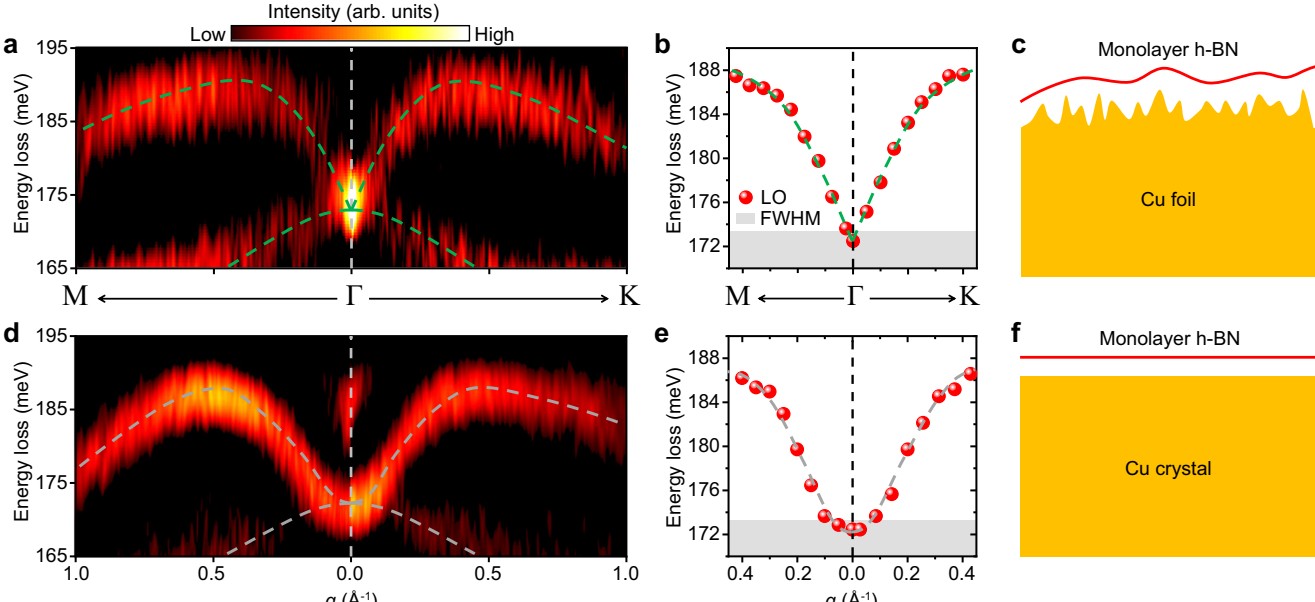

**Fig. 3 | Comparison of LO phonon of monolayer h-BN on Cu foil and Cu single crystal substrates. a** 2D phonon spectra of the LO phonon of h-BN/Cu foil. **b** LO phonon dispersion data of h-BN/Cu foil extracted from EDC. **c** A schematic of the morphology of the monolayer h-BN/Cu foil. **d**–**f** Results for monolayer h-BN/Cu crystal corresponding to **a**–**c**. The dashed curves in **a**, **b**, **d**, and **e** are guides to the eye. The height of the gray shaded areas in **b** and **e** represent the full width at half maximum (FWHM) of the zero-loss peak, reflecting the energy uncertainty of the measured data. Note that the weak intensity around 185 meV at the CBZ in **d** is contributed by the adsorption vibration of carbon[62], which does not affect the results, as discussed in detail in Supplementary Fig. S7 in the SI.

characteristics of the Cu foil substrate, we performed a comparative 2D-HREELS measurements on the phonons of h-BN/Cu crystal. The atomic-level flatness of Cu single crystals has been demonstrated by previous experiments[30]. In Fig. 3d, e, the second derivative phonon spectra and the extracted dispersion of h-BN/Cu crystal are shown (the original spectra in Supplementary Fig. S6 in the SI), providing a comparison with Fig. 3a, b of h-BN/Cu foil, respectively. It is evident that the dispersion of the LO phonon exhibits a "U-shaped" analytic behavior near the Γ point in the h-BN/Cu crystal. This suggests that the strong screening effect of the Cu crystal substrate completely suppresses the finite slope of the LO phonon at small momenta, resulting in the dispersion similar to that shown in Fig. 1a. The complete suppression of the nonanalytic behavior of LO phonon by the metal substrate is also observed in HREELS measurements of monolayer h-BN grown on Ni(111)[31]. However, the complete suppression does not occur in h-BN/Cu foil, as if the screening effect from Cu foil deviates from the commonly-known screening behavior of regular metals.

The screening effect observed in h-BN on a metal substrate primarily originates from quantum nonlocal response. The efficacy of nonlocal screening diminishes rapidly with the increasing separation between the metal substrate and the 2D monolayer, which has been experimentally demonstrated in graphene/metal system[32]. In the case of the h-BN/Cu foil system, we attribute the linear dispersion of LO phonons near the Γ point to the increased distance between the substrate and h-BN, a consequence of the surface roughness inherent in the Cu foil. As illustrated in Fig. 3c, f, when compared with a Cu single crystal substrate with an atomically flat surface, the surface roughness of the Cu foil substrate unavoidably increases the average distance between the monolayer h-BN and the substrate. Theoretically, calculations[33] affirm that when the distance between the metal substrate and monolayer h-BN exceeds 5 Å, the polarization electric field of h-BN can no longer be entirely screened by the nonlocal response. Additional evidence supporting this interpretation arises from the distance-dependent nature of the nonlocal screening. The distances between monolayer h-BN and Ni(111) and Cu(111) single crystal substrates are 2.1 Å[31] and 3.3 Å[34] respectively. In both instances, the linear dispersion is completely suppressed at momenta less than 0.18 Å$^{-1}$ for Ni(111) (ref. [31]) and 0.05 Å$^{-1}$ for Cu(111), respectively. All these facts underscore that the screening effect is extremely sensitive to the spatial separation between h-BN and the metal substrate. Although measuring the precise value of the average distance between h-BN and Cu foil is extremely challenging, it is undoubtable that the roughness of the Cu foil leads to a larger distance than 3.3 Å observed in the case of Cu(111) single crystal substrates. Consequently, it is not surprising that this increased distance results in an incomplete nonlocal screening.

## Properties of screened 2D PhPs

The observed breakdown of the LO-TO splitting fulfills the prerequisite for the existence of 2D PhPs in polar monolayers. As theoretically demonstrated, the LO phonons are simply equivalent to the 2D PhPs in polar monolayers under the premise of the breakdown of the LO-TO splitting[15]. Thus, information on the 2D PhP of monolayer h-BN can be directly derived from our measurements of the LO phonon dispersion. Two key figures of merit about the 2D PhPs in h-BN we investigate here are the deceleration factor and the confinement factor. The deceleration factor is a measure for the slowdown of light trapped in a PhP and is defined by the ratio of the group velocity of the PhP to the speed of free light, $\frac{v_g}{c} = \frac{1}{c}\frac{\partial\omega}{\partial q}$, where $\omega$ is the frequency of the PhP at a specific momentum $q$ and $c$ is the speed of light. The confinement factor is a measure for the compression of the wavelength of light trapped in a PhP and is defined by the ratio of the momentum of PhP to the wavevector of free light, $q/q_0$, where $q_0$ is the wavevector of free light at the corresponding PhP frequency. Figure 4a, b shows the comparison of the deceleration factors and confinement factors of h-BN for thick-layer (10 nm, data extracted from ref. [35]), freestanding monolayer (calculated results, Methods), and monolayer on Cu foil (our experimental results). The PhP in monolayer h-BN exhibits lower deceleration and larger confinement factors than the thick-layer. This can be easily deduced from the dependence of the dispersion of h-BN PhPs on the number of layers (Fig. 4c, Methods). Under a specific energy, thicker layers always have higher dispersion slopes and smaller

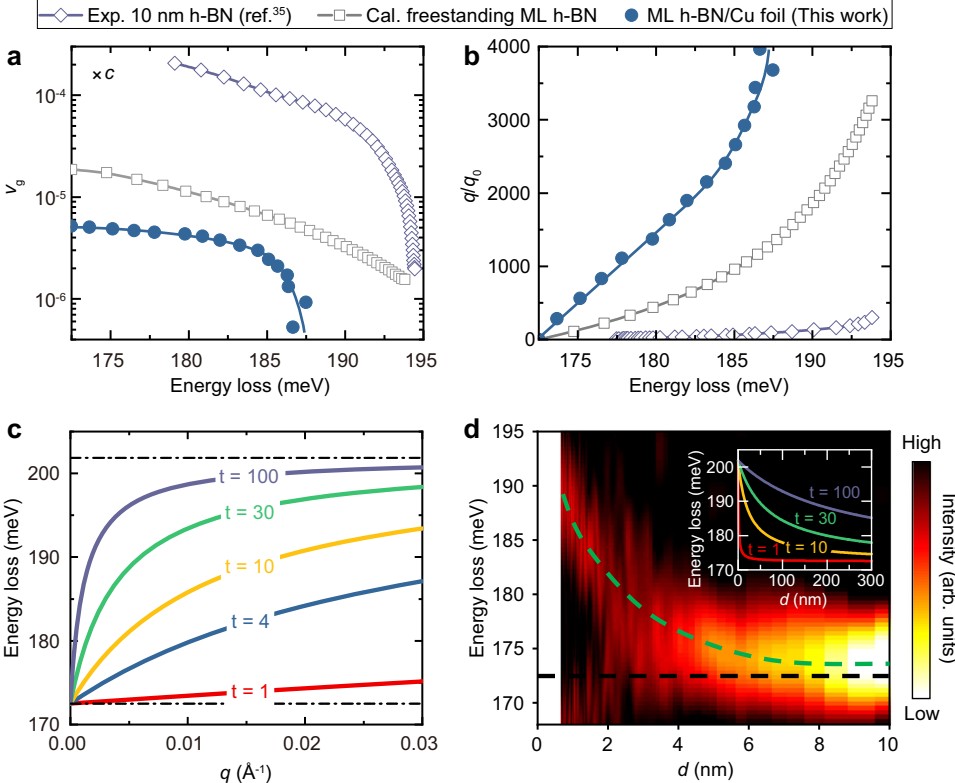

**Fig. 4 | 2D PhPs of monolayer h-BN. a, b** Deceleration and confinement factors versus energy loss, respectively. Solid circles represent the experimental results of this work, empty squares represent the calculated results for a freestanding monolayer (ML), and empty rhombus represent the measured results of 10-nm-thick extracted from ref. 35. The solid curves represent the fitting results to the corresponding data. **c** The calculated dispersion relation of the PhPs for various freestanding layers (Methods). *t* represents the number of layers of h-BN. **d** Spatial distribution of PhP transformed from 2D phonon spectra measured along Γ-K direction (0–0.4 Å⁻¹). Inset: calculated spatial distribution of PhPs for various layers using Eq. (1).

momenta than thinner layers, which gives the thinner layers a lower deceleration factor and higher confinement factor. In this regard, the PhP in monolayer h-BN has the optimal deceleration factor and confinement factor. Shockingly, our experimental results show that the PhP of monolayer h-BN on Cu foil has lower group velocity and higher confinement factor than the calculated results of the freestanding monolayer h-BN. The PhP of h-BN/Cu foil exhibits an ultra-slow group velocity down to about $5 \times 10^{-6}$ c at the long-wave limit and an ultra-high confinement factor up to about 4000, surpassing the existing reported PhP records of the lowest group velocity (~$10^{-5}$ c) and the highest confinement factor (~500) (ref. 35), respectively.

Although tuning the PhPs of thick-layer h-BN via the screening of metallic substrate has been extensively investigated[36–39], it has not been explored experimentally in strict monolayer h-BN. Computational analyses indicate that in the presence of a metal substrate, PhPs in a strict polar monolayer can undergo hybridization with their electromagnetic "mirror image", resulting in the emergence of the so-called acoustic PhP[33,40]. These acoustic PhPs exhibit larger confinement, stronger near-field enhancement, slower group velocities, and nearly identical polariton lifetimes when compared to conventional PhPs in freestanding monolayers[33]. In our study, the observed dispersion and properties of the 2D PhPs align with the characteristics predicted for acoustic PhPs, thanks to the nonlocal screening provided by the Cu foil substrate. Nevertheless, to conclusively establish whether the observed PhPs are indeed acoustic in nature, more high-precision experiments are needed in the future.

### Comparison of the methods for measuring PhPs

Currently, the mainstream methods for measuring PhPs are s-SNOM and STEM-EELS, both of which obtain the dispersion information of

PhPs through real-space measurements. The PhPs are excited by an incident beam (light or electron) near the sample boundary. The excited PhPs propagate to the edge, and are reflected by the edge. And then the reflected and the original PhPs interfere with each other, leading to a maximum interference intensity at

$$2qd + \varphi = 2\pi \qquad (1)$$

where $d$ is the distance to the edge boundary, $\varphi$ is the phase change introduced by the reflection[41,42]. Here we set $\varphi = \pi/4$ according to ref. 42. The dispersion of a PhP can be obtained by measuring the excitation energy and $d$ according to Eq. (1). We need to note that in Eq. (1), $q$ is inversely proportional to $d$. This makes the PhPs with very sharp dispersions instead exhibit very flat spatial distributions, and vice versa (see the dispersions and spatial distributions of PhPs for 100 layers and monolayer h-BN in Fig. 4c and the inset of Fig. 4d). In this sense, the tools that perform measurements in real-space and momentum-space are complementary in measured precision and range. In thick-layer samples, the sharp dispersion of PhPs at small momentum makes it challenging for 2D-HREELS measurements, but its flat spatial distribution enables s-SNOM/STEM-EELS to measure the dispersion of PhPs with high precision[18,20,35,41–44]. In contrast, in monolayer samples, the group velocity of the LO phonon is small and can be easily captured by 2D-HREELS, but the spatial distribution of 2D PhP is very sharp. As shown in Fig. 4d, converting our measured results to the spatial distribution of PhP near the h-BN sample boundary by Eq. (1), we can see that the sharp change of PhP energy with $d$ occurs within ~6 nm. Considering the strong localization and defects at the sample boundary[35] and the spatial resolution of the s-SNOM/STEM-EELS, it is extremely challenging to perform spatially

**Table 1 | Comparison of different experimental techniques for PhP measurements**

| | Raman | s-SNOM | STEM-EELS | 2D-HREELS |
|---|---|---|---|---|
| Detection type | Energy and momentum transfer in momentum-space | Standing wave in real-space | Standing wave in real-space | Energy and momentum transfer in momentum-space |
| Spatial resolution | ~30 μm[60] | ~10 nm[19] | ~0.2 nm[35] | ~1 mm |
| Momentum resolution | ~$10^{-7}$ Å$^{-1}$ (ref. [47]) | Better than $10^{-7}$ Å$^{-1}$ (ref. [18])[a] | Better than $10^{-7}$ Å$^{-1}$ (ref. [35])[a] | $10^{-3}$ Å$^{-1}$ |
| Energy resolution | ~0.1 meV[47] | ~$10^{-2}$ meV[61] | 5.5 meV[44] | ~1 meV |
| Momentum range | $0 \le q \le 10^{-3}$ Å$^{-1}$ (ref. [47]) | $0 < q \le 10^{-3}$ Å$^{-1}$ (ref. [19]) | $0 < q \le 10^{-2}$ Å$^{-1}$ (ref. [35]) | $0 \le q \le 2$ Å$^{-1}$ |

[a]The momentum resolution of s-SNOM and STEM-EELS are converted from the distributions of standing waves in real space. These values are sample dependent. Here, we cite the typical values from the measurements of h-BN.

resolved measurements of 2D PhPs within such small $d$. In this regard, 2D-HREELS provides a new methodology for investigating the properties of 2D PhPs in a large momentum range.

It is worth noting that STEM-EELS has also demonstrated the ability to measure the phonon dispersion of thick-layer h-BN[45,46], but it is insensitive to the monolayers and has lower energy resolution, making it difficult to capture phonon dispersion in strictly monolayer h-BN. We also noticed that a recent study using Raman spectroscopy with backscattering configuration[47] can measure PhPs of thick (200-750 nm) 2D materials. A comparison of different experimental techniques for PhPs measurements is listed in Table 1.

## Outlook of the measured phonon and polariton behavior

Through precise measurements of the phonon spectra in monolayer h-BN with high energy and momentum resolutions, our study systematically explores the behaviors of the LO phonon from the CBZ to a large momentum range. The degeneracy of LO and TO phonons, as well as the "V-shaped" nonanalytic behavior of the LO phonon at the CBZ, are comprehensively demonstrated in our 2D spectra. Notably, our investigation reveals that the screening from the Cu foil substrate reduces the slope of the LO phonon. This results in the 2D PhP of h-BN exhibiting an ultra-slow deceleration factor (-5 × $10^{-6}$ $c$) and ultra-high confinement factor (-4000) surpassing the 2D limit of freestanding h-BN. This advancement is expected to facilitate further development of subdiffraction imaging[48–50], nanoresonators[41,51], and single molecule detection[40], among others[52]. We also emphasize the complementarity of measuring PhPs in both real-space and momentum-space, introducing a new methodology for scrutinizing the physical properties of 2D PhPs through 2D-HREELS.

## Methods

### h-BN/Cu foil preparation

A polycrystalline Cu foil substrate with a thickness of 25 μm (sourced from Sichuan Oriental Stars Trading Co. Ltd) was employed in the synthesis of monolayer h-BN film. Prior to growth, the Cu foil was subjected to a heat treatment in a tube furnace, where it was heated to 1050 °C over a period of one hour and annealed for an additional hour in a mixed gas flow of Ar (500 sccm) and H$_2$ (50 sccm) at atmospheric pressure. The precursor of ammonia borane with 97% purity (sourced from Aldrich) was placed in a ceramic crucible situated 1 m upstream from the substrate in the heating zone of the tube furnace. The system was then switched to low pressure (approximately 200 Pa) and subjected to a mixed gas flow of Ar (10 sccm) and H$_2$ (40 sccm). The precursor was heated to 65 °C within a period of 10 min and held at this temperature for 1.5 h using a belt heater, allowing for sublimation and the subsequent growth of h-BN film on the substrate. After growth, the system was cooled naturally to room temperature under a gas flow consisting of Ar (500 sccm) and H$_2$ (10 sccm) at atmospheric pressure.

### h-BN/Cu crystal preparation

Fabrication of single crystal Cu(111) film. The 500-nm-thick Cu film was deposited on a single-crystal sapphire (4 in., c plane with misorientation

<0.5°, 600 μm thickness, Epi-ready with R$_a$ < 0.2 nm) by sputtering using a physical vapor deposition (PVD) equipment (ULVAC, QAM-4W). A DC power of 200 W was used to deposit Cu at 20 nm/min. Then, the Cu/sapphire was annealed at 1000 °C with 500 sccm Ar and 100 sccm H$_2$ at atmospheric pressure for 1 h, giving rise to a single-crystal Cu(111)/sapphire wafer.

Growth of h-BN on Cu(111)/sapphire. The monolayer h-BN films were grown by chemical vapor deposition (CVD) using a tube furnace (Thermal Scientific). The precursor was about 60 mg ammonia borane (97%, Sigma Aldrich), which was filled into a quartz tube and placed in the upstream side of the growth chamber. The Cu(111)/sapphire was heated to 1000 °C with 500 sccm Ar and 100 sccm H$_2$ at atmospheric pressure. Then the CVD system was switched to 50 Torr with 3000 sccm H$_2$, while the precursors were heated to 60 °C by a heating belt and maintain 10 min. After h-BN growth, the sample was cooled rapidly to room temperature.

### HREELS measurements

In a HREELS measurement, the energy ($E_{loss}$) and momentum ($q$) of the phonons are obtained by the conservation of energy and momentum for the incident and scattered electrons. As given by

$$q = \frac{\sqrt{2mE_i}}{\hbar} \sin\theta_i - \frac{\sqrt{2mE_s}}{\hbar} \sin\theta_s \tag{2}$$

and

$$E_{loss} = E_i - E_s \tag{3}$$

where $\theta_i$ ($\theta_s$) and $E_i$ ($E_s$) are the incident (scattered) angles and energies of electrons, respectively. The conventional HREELS collects the energy loss curves of scattered electrons at a fixed angle in each measurement and the dispersion relation is achieved by rotating the monochromator, analyzer, or sample. Our developed 2D-HREELS can directly obtain a 2D energy-momentum mapping simultaneously by a specially designed lens system with a double-cylindrical Ibach-type monochromator combined with a commercial VG Scienta hemispherical electron energy analyzer[22]. A comparison of 2D-HREELS and conventional HREELS is shown in Supplementary Fig. S1 in the Supplementary Information. With this setup, the ultimate energy and momentum resolutions are better than 0.7 meV and 0.002 Å$^{-1}$ (ref. [22]). To compromise with the detection efficiency, we have set the resolutions to 3.3 meV and 0.025 Å$^{-1}$ for energy and momentum, respectively.

The HREELS and LEED are equipped in an ultrahigh-vacuum system with a base pressure better than 2 × $10^{-10}$ Torr. After being transferred to the 2D-HREELS system, the samples were annealed at 400 °C to remove possible contamination. The measurement direction of the samples is precisely controlled by the electric motors and examined by the LEED.

2D-HREELS data were processed in the commercial software Igor Pro 9. The second-derivative spectra are represented in negative second derivative $-d^2I/d\omega^2 > 0$, where the $I$ is the intensity and $\omega$ is the energy loss. In order to check the reliability and reproducibility of the

data, we also carried out 2D-HREELS measurements on additional samples (Supplementary Fig. S7). The measurement data reproduced the results shown in the text very well.

## STM measurements

The STM experiments were performed after HREELS measurements. The sample was transferred ex-situ into the ultra-high vacuum STM system (Unisoku) with a base pressure better than $2 \times 10^{-10}$ Torr. After being transferred to the STM system, the sample was also annealed at 400 °C. The morphology characterization of h-BN/Cu was performed at 78 K. Pt/Ir tips were used in the STM experiments.

## Theoretical calculation

The phonon dispersion of monolayer h-BN was calculated using the DFPT[53] implemented in the Quantum ESPRESSO[54,55]. In traditional DFPT processing, the spurious interactions introduced by the periodic boundary conditions imposed on polar 2D materials cannot correctly calculate the LO phonon dispersions at small momentum. To get the correct dispersion of the LO phonon, we performed the 2D implementation of DFPT developed by Sohier et al.[12,27,28]. We use the Perdew-Burke-Ernzerhof (PBE) exchange-correlation functional[56] combined with the Projector Augmented Wave (PAW) pseudopotentials[57]. A 55 Ry kinetic energy cut-off, a $12 \times 12 \times 1$ electron momentum grids, and an $8 \times 8 \times 1$ phonon momentum grids are applied. The optimized lattice parameters of h-BN are 2.509 Å, which shows excellent consistency with the earlier works[58,59].

The PhP dispersions of freestanding h-BN with different layers are calculated according to the following model[12]

$$\omega_{\mathrm{LO}}^{t}(q) = \sqrt{\omega_{\mathrm{TO}}^2(q=0) + t\, S\, q/(1+t r_{\mathrm{eff}}\, q)} \qquad (4)$$

where $\omega_{\mathrm{LO}}^{t}(q)$ is the LO phonon frequency at momentum $q$, $t$ is the number of layers, $\omega_{\mathrm{TO}}(q=0)$ is the TO phonon frequencies at $q=0$, $S$ is a parameter related to the Born effective charges and phonon displacements, $r_{\mathrm{eff}}$ is the effective screening length describing the screening properties of the 2D material itself. From our measurement results, we have $\omega_{\mathrm{TO}}(q=0)=172.5\,\mathrm{meV}$. $S$ and $r_{\mathrm{eff}}$ are material-specific properties independent of the surrounding environment. Here, we set $S=8.40\times10^{-2}\,\mathrm{eV}^2 \cdot \mathrm{Å}$ and $r_{\mathrm{eff}}=7.64\,\mathrm{Å}$, as obtained from the calculations with density functional theory[12].

## Data availability

All data generated or analyzed during this study are included in this published article (and its Supplementary Information files). The HREELS data generated in this study have been deposited in the figshare database under accession code https://doi.org/10.6084/m9.figshare.25239607. All additional information is available from the corresponding authors upon request.

## Code availability

The numerical code used in this work, Quantum ESPRESSO, is open-source and can be accessed at: https://www.quantum-espresso.org/.

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

## Acknowledgements

This work was supported by the National Key R&D Program of China (Grants No. 2021YFA1400200, X.Z.; No. 2022YFA1403000, X.Z.), the National Natural Science Foundation of China (Grant No. 12274446, X.Z.), and the Strategic Priority Research Program of the Chinese Academy of Sciences (Grant No. XDB33000000, J.G.). X.Z. was partially supported by the Youth Innovation Promotion Association of Chinese Academy of Sciences.

## Author contributions

J. L. and Z. T. performed the HREELS experiments. J. L. performed the theoretical calculation and analysis. L. W. grew the monolayer h-BN sample on Cu foil. Y. W. and H. P. grew the monolayer h-BN sample on Cu single crystal. W. W., W. Z., and G. M. performed the STM experiments. S. X. and Z. S. participated in the data analysis and discussion. J. L., X. Z., and J. G. wrote the manuscript with input from all authors. X. Z. and J. G. supervised the project. All authors contributed to the discussion of the results.

## Competing interests

The authors declare no competing interests.
