## [Peer Review File · Nature Communications]

Reviewers' Comments:

Reviewer #1:

Remarks to the Author:

In this work, the authors use a fairly recent technique (2D-HREELS) to study the phonon dispersion of monolayer BN. In particular, they observe distinctive features in the dispersion of polar-optical phonons (i.e., LO-TO splitting) that were theoretically predicted to be signatures of the 2D nature of the material. Namely, LO and TO modes are degenerate at zone center, then the LO dispersion increases linearly at small momenta, with a discontinuity in the first derivative of the LO dispersion at zone center.

The development of spectroscopic techniques with enough energy and momentum resolution for a full, accurate, and direct characterisation of the phonon dispersion of monolayer 2D materials would be a major breakthrough in the field. It would unlock further experimental study of these systems with various fundamental and technological applications, such as polaritonics.

Different techniques are going in this direction (s-SNOM, STEM-EELS, 2D-HREELS, ...). The authors provide an overview of recent progress and argue the strengths of their developments for 2D-HREELS.

The manuscript is well written and clear. The direct measurement of the optical phonon dispersion in a monolayer with such resolution in the entire Brillouin zone is a clear step forward in the field. The authors make a convincing demonstration of their experimental technique in general. Concerning the observation of the 2D LO-TO splitting, however, some of the findings are quite puzzling and should be addressed in a revised manuscript before a final decision can be reached.

My main concern is that the linear increase of the LO dispersion at small momenta should not be observed for BN on a metallic substrate. As far as I understand, this has been one of the major issues with the observation of this phenomenon in the past (e.g., Ref 43). The authors do address the screening from the environment, a key factor in the 2D LOTO mechanism. However, they arrive at a constant effective screening function of 4.45, when one would expect a metallic behavior, with ϵ^{-1} in $o(q)$, suppressing the finite slope at Gamma and making it at least quadratic. This reasoning assumes that the relevant energy and momentum range (0.16 to 0.2 eV and 0. to 0.5 A⁻¹) is under the dispersion of plasmons in the Cu foil and the momenta are smaller than the size of the Fermi surface of Cu. These seem like reasonable assumptions in Cu (or other common metals). The bulk plasma frequency should be around 7-8 eV, and potential acoustic surface plasmon would cross the LO dispersion around 0.05A⁻¹, using the slope from K. Pohl et al. 2010 EPL 90 57006.

Of course, this is "non-local" screening, becoming less efficient as the distance with the Cu foil increases since the induced fields decay as $\exp(-qz)$ from the Cu surface. However, the range of momenta relevant here should be small enough compared to the inverse distance between BN and the Cu foil, meaning the screening should be very efficient.

Whether some part of the reasoning above is flawed, or some peculiarities of the experimental setup (surface roughness of Cu foil?) might invalidate some parts of it, this issue should be addressed in any case.

Also, and maybe related to the above, the TO dispersion seems to be presenting a finite negative slope around zone center. This is not predicted by theory and should also be addressed.

Other smaller issues or comments are listed below:

- it might be argued that the definite observation of the breakdown at zone center was previously achieved using Raman spectroscopy in the paper: "Experimental demonstration of the suppression of optical phonon splitting in 2D materials by Raman spectroscopy", Marta De Luca et al. 2020 2D Mater. 7 035017. The observation of the finite slope at small momenta however, is much more novel, even if partly or indirectly seen in some previous experiments, as indicated by the authors (e.g. Ref. 33)

- There is a slight confusion about 2D DF(P)T calculations (e.g. p. 4 l. 93, and in the methods). The obtention of the proper dispersion requires more than "the 2D implementation of the non-analytical term correction". The phonon calculations are first done on a coarse grid of q-points. Already at this point, the code is modified to implement 2D periodic boundary conditions and give

the correct phonon properties on this coarse grid. Then, the Fourier interpolation of this coarse phonon dispersion indeed requires to deal with non-analytical contributions that are different in 2D (with respect to the well known 3D ones). One might simply say something like "the phonon dispersions are calculated with the 2D implementation of density functional perturbation theory". - Related to the above, the gray curve of Fig. 3a is the result of computing the 2D or 3D phonon frequencies (difficult to say which here) on the coarse grid, but without the proper correcting terms in the interpolation. In a way, it is somewhat not self-consistent. It might make more sense here to compare the full, correct 2D treatment (the green curve) with a full 3D treatment (with the correction 3D non-analytical terms in the interpolation, like the blue curve in Fig 2 of Ref. 12). The latter is still physically wrong, as it includes spurious effects from the periodic images, but at least the direct coarse-grid calculations and the interpolation are treated consistently, in the same dimensionality framework.

Reviewer #2:

Remarks to the Author:

The manuscript analyzes the phonons in a monolayer of hexagonal boron nitride (h-BN) with the help of high-resolution electron energy-loss spectroscopy (HREELS). Based on the measured data, the authors confirm the disappearance of the LO-TO splitting at the Gamma point in 2D monolayers of polar materials and support the theoretical predictions (e.g., Ref. [12]). Similar findings have however been made even experimentally (Ref. [43]), so probably the most exciting result is the high confinement of the optical phonon polaritons (PhPs), most likely introduced due to the copper substrate below the h-BN monolayer.

While I find the manuscript well-organized and easy to follow, there are still several important points that should be clarified:

1. The polariton dispersion is modeled with the help of Eq. (1). The results of the fitting, where the authors find the dielectric response of the copper substrate $\epsilon_{\text{bot}}=4.45$, however, seem to be in conflict with the fact that copper should exhibit metallic behavior in the frequency region of interest. That questions either the validity of the model, or values of other parameters (S , r_{eff}), which are extracted for a free-standing monolayer and might not be valid for a monolayer on top of metal. The copper substrate might introduce strain, screening, and non-local effects at the atomistic level or yield the emergence of the so-called mirror/acoustic polaritons, where the polaritons in h-BN hybridize with their "mirror image" if there is a metal underneath. The role of the substrate might be rather non-trivial, so it could be crucial to perform the study with different substrates.

2. Possibly related to the previous point: based on Fig. 3, the values of $\omega_{\text{(TO/LO)}}$ near $q=0$ found from the theoretical modeling do not match the experimental results very nicely.

3. The authors emphasize some advancements they made to improve the HREELS setup compared to conventional arrangements. I am missing a schematic of their improvements. They should also mention the energy-momentum resolution achievable with their setup and comment on the data processing (e.g., extraction of the second-derivative spectra).

Other remarks:

4. Introduction, lines 27 – 29: this sounds as over-claiming. This study only covers phonon dispersions in one representative 2D polar material, so I cannot see how this can be universal.

5. 2D color plots are missing the color bars with the measured quantities, their values, and units.

6. Fig. 2 a,b show rather nonuniform intensities – this should be commented.

7. Fig. 2d: lines denoting phonon replicas are hardly visible.

8. What is the meaning of vertical dashed lines in Fig. 3a?

9. As I understand, the gray area in Fig. 3b shows uncertainty in E- resolution. This should be explicitly mentioned as it also imposes uncertainties for the extracted points in Fig. 3b.

10. Lines 143-144: there are studies with free-standing or encapsulated monolayers, so semi-infinite substrates are not the only option for applications.

11. Line 180: in literature, v_g typically denotes the group velocity, not its value normalized by c .

12. Line 221: SETM should be STEM.

13. Fig. 4 a,b: I anticipate that the dark blue lines are extracted from the fit, while the dark blue dots are from the presented measurement – this should be explicitly mentioned.

14. The last section already discusses the comparison with other techniques for measuring PhPs. It would also be nice to explicitly state an effective q/d -range + q/d -resolution of the different techniques (HREELS, s-SNOM, STEM-EELS) in a table. Also, besides Refs. [19] and [33] dealing with PhPs in STEM-EELS, there is a more recent reference, which might be mentioned [Small 17, 2103404 (2021)].

15. Methods section dealing with HREELS could also provide references to a more detailed theory behind the technique.

Reviewer #3:

Remarks to the Author:

Review of "Observation of the Breakdown of Optical Phonon Splitting in a Two-dimensional Polar Monolayer" by Li et al.

In this work, Li and coauthors present direct evidence that in a monolayer of hexagonal Boron Nitride, the longitudinal optical (LO) and transverse optical (TO) phonons at the zero center exhibit no energy splitting, contrary to the conventional LO-TO splitting seen in 3D materials. Despite extensive theoretical descriptions of this behavior, the authors claim this to be the first experimental verification of the LO-TO splitting breakdown. They then characterize the LO phonon (or, if you like, the phonon polariton) on a Copper foil substrate and describe the effect of screening on the momentum dependence near the zone center, along with its deceleration and confinement factors.

The work is interesting. To be honest I'm not sure how important the work is for the broader scientific community, as I will try to explain. The research appears to be thorough and claims are generally well-founded. I provide more detailed comments below that may help the authors improve their manuscript further.

1) The authors claim that this is the first experimental observation of the absence of LO-TO splitting. They point to ref. 20 and say "a weak 2D PhP signal has been detected...the momentum window is too narrow to reveal the overall properties of the 2D PhPs." This reference appears to observe LO phonons in monolayer hBN having linear dispersions near the Brillouin zone center, consistent with this work. While I understand that the experimental measurement technique is different, why is it fair to claim that the result presented here is the first experimental observation of the absence of LO-TO splitting? It's quite clear from figure 1 and also from theory that the TO phonon should not be impacted by the dimensionality of the Coulomb interaction. Thus measuring the LO phonon frequency to be that near the TO frequency (at finite wavevector) all but guarantees that LO-TO splitting (at the Brillouin zone center) is absent. I wonder about how differentiable your results are from ref. 20 from the standpoint of verification of no LO-TO splitting.

2) Theoretical works cited in this manuscript have in some sense extensively studied the expected impact of a 2D Coulomb interaction on the LO-TO optical phonon dispersions. Since 2019, we have known that for polar 2D systems: 1. there should be no LO-TO splitting at the Gamma point, 2. the LO phonon should have a linear slope near the Gamma point and 3. the LO phonon is essentially a

phonon polariton. All three of these conclusions fall out of the 2D nature of the Coulomb interaction in these monolayer or effectively aperiodic-in-z systems. Reading this paper, none of the results shown in figs. 1-3 were at all surprising.

My question is this. Why do the authors claim that this work is exceptionally useful? For example they state "Combined with first-principles calculations, it is demonstrated that this exotic behavior originates from the long-range Coulomb interactions of lattice vibrations." This statement is true, but it's hardly a result of this work. It has been known for years. They also state "Our experimental results demonstrate the correctness of the 2D implementation of the nonanalytical-term correction and lay the foundation for a correct understanding of the behavior of the LO phonons in 2D polar monolayers." Again, I find this misleading. The correct understanding of the behavior of the LO phonons in 2D polar monolayers has been known by the community for at least four years. One can argue that experimental verification was already provided in ref. 20 four years ago as well. Various follow up works involving phonon polaritons in 2D systems have been carried out since then. How can this paper then claim to lay the foundation of this field?

3) Following up on this, I don't find the theoretical calculations in this work particularly useful. In fact, they are in many ways directly reproducing those from ref. 12. Figure 3a in this manuscript is hardly different qualitatively from the comparison done in figure 2 in ref. 12. The authors use figure 3 to state that "This demonstrates that the long-range Coulomb interaction clearly affects the dispersion of the LO phonons in 2D polar materials." Why is this statement presented in such a way that it is a new finding? This is exactly the finding presented in ref. 12! I do not understand why the authors do not, at the very least, give proper credit for this finding to ref. 12, since this exact phenomena has been explained so extensively in their work. Instead, the current text makes it sound as if the authors were the first to discover this result.

4) In the fit presented in fig. 3b, the authors find that the effective substrate screening is 4.45. Can the authors contextualize what this means? The authors make it a point that the Cu foil used is a good substrate for modifying the polariton dispersion without breaking the induced electric field. Can the authors comment on what makes Cu foil good, and/or what material properties would be necessary to optimize substrate interactions for a specific dispersion or polariton property? Including this kind of analysis, and if possible any more experimental data with different substrates, to improve the manuscript's novelty quite a bit.

5) The authors conclusion includes idea that "monolayer h-BN/Cu foil is an ideal system for exploring light-matter interactions." I don't really understand what this means. "exploring light-matter interactions" is an extremely vague statement. What do the authors really mean? There are plenty of aspects about these polaritons that are never discussed, for example their expected lifetimes, which would greatly impact their potential for different light-matter-based applications. Can the authors please be more specific with their outlook.

Reviewer #4:

Remarks to the Author:

This work provides experimental evidence, supported by DFPT calculations, of the breakdown of the TO-LO splitting in a monolayer of h-BN, used as a prototype of polar 2D materials. There are some strong points and some weak points that I discuss below.

The strong points of this work are:

- Challenging 2D-HREELS measurements providing directly a 2D energy-momentum mapping of monolayers; this is the first experimental measurement of the 2D phonon dispersion of monolayers
- Effect of the Cu foil substrate is new: finding of the ultra-slow group velocity and ultra-high confinement is unexpected
- Methods, data analysis, calculations, interpretation and conclusions are sound
- Very well written manuscript and high scholar presentation

- Comparison of the methods for measuring phonon polaritons in the last paragraph is useful and appealing to a broad audience

The weak points are:

- the paper is entirely based on one measurement only (although very good), which is replotted in several different ways, but it is always the same measurement. The new data are only in figure 2c. Then figure 2d and 2e are extracted from that (which is normal and necessary), and figure 3a and 3b are zooms of the same data (again, this is normal and necessary), and figure 4 contains again the same data, with the comparison with literature (again, this is good). The SI do not show any new data, as well. From the discussion of the data I can imagine that with this technique the authors cannot study any dependence on the layer thickness nor on the type of substrate, but there are studies that should have been performed and could have been shown at least in the SI: the same measurement could have been performed on different materials or at least on different samples of the same material, even fabricated in the same way. The lack of any additional measurement poses a problem of reproducibility.

- the noteworthy results are the demonstration of the breakdown of the TO-LO splitting in a monolayer of h-BN and the effect of the substrate; while the latter is new, it is the first one that is the most relevant for future studies, and unfortunately it is not entirely new, as it has already been demonstrated at gamma point. Indeed, the paper 'Experimental demonstration of the suppression of optical phonon splitting in 2D materials by Raman spectroscopy' published in 2D materials in 2020 has basically the same title of this paper (and the same introduction and motivation), but it is not discussed. That paper concerns Raman measurements of WSe₂ and h-BN (two different materials), and it also studies the thickness dependence and the dependence on the type of stacking. Although the work from Jiade Li et al. is more complete (and more difficult experimentally) because thanks to HREELS also measures phonons far from gamma, its aim is still to demonstrate the correctness of the theory in ref. 12, which has already been demonstrated at least at gamma by the other paper. This poses a problem of novelty.

- the DFPT calculations are useful and well performed, but they reproduce calculations that are very similar to those in ref.12, thus the main novelty of this paper remain the experimental finding (with the limitations just discussed)

- there is some 'overselling' in some parts of the manuscript. Examples:

- the title refers to a 2D polar monolayer and suggests a universal finding, but the paper is only about h-BN. Although there is no reason why the results should not be applicable to other monolayers, I suggest writing h-BN in the title, otherwise the reader will think that more materials are measured
- in the conclusions, 'Combined with first-principles calculations, it is demonstrated that this exotic behavior originates from the long-range Coulomb interactions of lattice vibrations' is not completely correct, because this has already been demonstrated both theoretically and experimentally. I would rather write 'Our work provides additional experimental verification of the fact that this exotic behavior originated from, as already been predicted in ...'.

- Can the authors try to explain why the signal intensity of the TA mode is very weak? Is this a general finding in other materials studied with this technique? Is this expected?

- line 120: remove 's' from exhibits

- the discussion of the possible applications of these findings should be slightly expanded. Now it is quite vague and not appealing to a broad audience: which are the devices and the applications (and the fundamental studies) that would benefit from the strong light matter interaction?

In conclusion, I believe that the paper is very good and highly significant and as such worthy of being published in a very good journal. However, it lacks the necessary novelty, experimental reproducibility, and comparison with existing literature, necessary to be published in this high impact journal.

Point-by-point Response to Reviewers' Comments

We appreciate the reasonable judgments of the manuscript and the instructive suggestions raised by the reviewers. According to them, we have added comparative experiments of the monolayer *h*-BN on Cu single crystal substrates and revised the statement extensively. With these new inputs and revisions, we believe that all the comments have been addressed properly. The reviewers' comments are summarized in in blue italic font. Our detailed point-by-point response are in regular black font. The revision in the manuscript are quoted in black italic font.

We submit two revised versions: the clean version, and the highlighted version with all the changes marked to aid the reviewers in further review.

Reviewer #1 (Remarks to the Author):

Comment[#1-1]: *In this work, the authors use a fairly recent technique (2D-HREELS) to study the phonon dispersion of monolayer BN. In particular, they observe distinctive features in the dispersion of polar-optical phonons (i.e., LO-TO splitting) that were theoretically predicted to be signatures of the 2D nature of the material. Namely, LO and TO modes are degenerate at zone center, then the LO dispersion increases linearly at small momenta, with a discontinuity in the first derivative of the LO dispersion at zone center.*

The development of spectroscopic techniques with enough energy and momentum resolution for a full, accurate, and direct characterisation of the phonon dispersion of monolayer 2D materials would be a major breakthrough in the field. It would unlock further experimental study of these systems with various fundamental and technological applications, such as polaritonics.

Different techniques are going in this direction (s-SNOM, STEM-EELS, 2D-HREELS, ...). The authors provide an overview of recent progress and argue the strengths of their developments for 2D-HREELS.

The manuscript is well written and clear. The direct measurement of the optical phonon dispersion in a monolayer with such resolution in the entire Brillouin zone is a clear step forward in the field. The authors make a convincing demonstration of their experimental technique in general.

Reply[#1-1]: We would like to thanks for your positive comments about the importance and high quality of our work.

Comment[#1-2]: *Concerning the observation of the 2D LO-TO splitting, however, some of the findings are quite puzzling and should be addressed in a revised manuscript before a final decision can be reached.*

My main concern is that the linear increase of the LO dispersion at small momenta

should not be observed for BN on a metallic substrate. As far as I understand, this has been one of the major issues with the observation of this phenomenon in the past (e.g., Ref 43). The authors do address the screening from the environment, a key factor in the 2D LOTO mechanism. However, they arrive at a constant effective screening function of 4.45, when one would expect a metallic behavior, with epsilon-1 in $o(q)$, suppressing the finite slope at Gamma and making it at least quadratic. This reasoning assumes that the relevant energy and momentum range (0.16 to 0.2 eV and 0. to 0.5 \AA^{-1}) is under the dispersion of plasmons in the Cu foil and the momenta are smaller than the size of the Fermi surface of Cu. These seem like reasonable assumptions in Cu (or other common metals). The bulk plasma frequency should be around 7-8 eV, and potential acoustic surface plasmon would cross the LO dispersion around 0.05 \AA^{-1} , using the slope from K. Pohl et al. 2010 EPL 90 57006.

Of course, this is "non-local" screening, becoming less efficient as the distance with the Cu foil increases since the induced fields decay as $\exp(-qz)$ from the Cu surface. However, the range of momenta relevant here should be small enough compared to the inverse distance between BN and the Cu foil, meaning the screening should be very efficient.

Whether some part of the reasoning above is flawed, or some peculiarities of the experimental setup (surface roughness of Cu foil?) might invalidate some parts of it, this issue should be addressed in any case.

Reply[#1-2]: We appreciate your insightful comments, which was not clearly stated in the original version. The screening effect of the substrate is indeed critical.

To address this, we have conducted comparative experiments involving monolayer *h*-BN grown on an atomically flat Cu single crystal substrate. Our experimental results demonstrate that the metallicity of the Cu crystal substrate induces a significant screening effect, resulting in the LO phonon of *h*-BN exhibiting a parabolic form near the zone center. The only distinction between Cu foil and Cu single crystal lies in the surface roughness. The surface roughness unavoidably increases the distance between the Cu foil substrate and *h*-BN. Through this comparison, we attribute the linear dispersion behavior to the weakening of the screening effect with increasing distance.

It has been theoretically predicted [ACS Photonics 7, 2610-2617 (2020)] that when the distance between monolayer *h*-BN and the metal substrate is 5 \AA , the substrate is unable to fully screen the polarization electric field of *h*-BN, consequently generating acoustic phonon polaritons with linear dispersion. Furthermore, this study also notes that the sensitivity of the screening effect to distance changes is manifested in the roughness of the metal substrates. Considering that the distance between *h*-BN and the atomically flat Cu single crystal is about 3.3 \AA , it is not surprising that the additional increase in distance resulting from the surface roughness of the Cu foil leads to incomplete nonlocal screening.

Changes

We have accordingly made corresponding modifications in the "Screening effect from

Cu foil” section and Fig. 3 of the main text.

Comment[#1-3]: *Also, and maybe related to the above, the TO dispersion seems to be presenting a finite negative slope around zone center. This is not predicted by theory and should also be addressed.*

Reply[#1-3]: In fact, the momentum position presenting a finite negative slope is already larger than 0.1 \AA^{-1} . The negative slope dispersion of TO phonons under large momentum is consistent with the calculated results. At small momentum, the signal intensity of TO phonons is weak, and the energy is close to that of LO phonons, thus our experiments cannot obtain the dispersion results of TO phonons near the zone center. However, comparing the experimental results on Cu foil and Cu single crystal substrates (Figs. 3a and 3d), TO phonons exhibit the same negative dispersion behavior at large momentum, indicating that their behavior is not affected by nonlocal screening.

To address potential misunderstanding, we have added guidance lines for TO phonon dispersion in Figs. 3a and 3d.

It is essential to emphasize that the second derivative spectra accurately depict the “V-shaped” dispersion behaviors of the LO phonon near the zone center, owing to the prominent intensity of LO mode. This is demonstrated by our extraction of fitted LO phonon dispersion data (Fig. 3b).

Comment[#1-4]: *Other smaller issues or comments are listed below:*

*- it might be argued that the definite observation of the breakdown at zone center was previously achieved using Raman spectroscopy in the paper: "Experimental demonstration of the suppression of optical phonon splitting in 2D materials by Raman spectroscopy", Marta De Luca _et al_ 2020 _2D Mater._ **7** 035017. The observation of the finite slope at small momenta however, is much more novel, even if partly or indirectly seen in some previous experiments, as indicated by the authors (e.g. Ref. 33)*

Reply[#1-4]: Thank you for providing the important reference. In the revised version, we have modified the statement and added a discussion of this reference.

Changes:

Page 3, line 56: “Raman scattering has revealed the degeneracy of LO and TO phonons at the CBZ¹⁷, but it fails to detect the linear dispersion of LO phonons due to the lack of momentum resolution.”

Comment[#1-5]: *- There is a slight confusion about 2D DF(P)T calculations (e.g. p. 4 l. 93, and in the methods) . The obtention of the proper dispersion requires more than "the 2D implementation of the non-analytical term correction". The phonon*

calculations are first done on a coarse grid of q-points. Already at this point, the code is modified to implement 2D periodic boundary conditions and give the correct phonon properties on this coarse grid. Then, the Fourier interpolation of this coarse phonon dispersion indeed requires to deal with non-analytical contributions that are different in 2D (with respect to the well known 3D ones). One might simply say something like "the phonon dispersions are calculated with the 2D implementation of density functional perturbation theory".

Reply[#1-5]: Thank you for your correction. Following the suggestion, we have revised the statements in the revised version. Additionally, we emphasize that the original work comes from Sohier *et al.*

Changes:

Page 4, line 102 "To assign the phonon modes measured by 2D-HREELS, the calculated phonon dispersions, with the 2D implementation of the density-functional perturbation theory (DFPT) developed by Sohier et al.^{12,27,28}, are also superimposed in Fig. 2c and 2d (green curves, see Methods for details)."

Page 16, line 376 (Method): "To get the correct dispersion of the LO phonon, we performed the 2D implementation of DFPT developed by Sohier et al.^{12,27,28}."

Comment[#1-6]: - *Related to the above, the gray curve of Fig. 3a is the result of computing the 2D or 3D phonon frequencies (difficult to say which here) on the coarse grid, but without the proper correcting terms in the interpolation. In a way, it is somewhat not self-consistent. It might make more sense here to compare the full, correct 2D treatment (the green curve) with a full 3D treatment (with the correction 3D non-analytical terms in the interpolation, like the blue curve in Fig 2 of Ref. 12). The latter is still physically wrong, as it includes spurious effects from the periodic images, but at least the direct coarse-grid calculations and the interpolation are treated consistently, in the same dimensionality framework.*

Reply[#1-6]: We have added results from calculated phonon dispersion using full 3D treatment and removed the results without the proper correcting terms. In order to emphasize the experimental comparison of different substrates, we have removed the calculated curves from Fig. 3 of the main text. *The new calculated results are presented in the Supplementary Information ("Comparison of calculated optical phonon dispersions" section and Fig. S5) to demonstrate the comparison between experiments and calculations.*

Reviewer #2 (Remarks to the Author):

Comment[#2-1]: *The manuscript analyzes the phonons in a monolayer of hexagonal boron nitride (h-BN) with the help of high-resolution electron energy-loss spectroscopy (HREELS). Based on the measured data, the authors confirm the disappearance of the LO-TO splitting at the Gamma point in 2D monolayers of polar materials and support the theoretical predictions (e.g., Ref. [12]). Similar findings have however been made even experimentally (Ref. [43]), so probably the most exciting result is the high confinement of the optical phonon polaritons (PhPs), most likely introduced due to the copper substrate below the h-BN monolayer.*

Reply[#2-1]: We appreciate your very detailed comments on our manuscript. With regards to the observation of the exotic behavior of LO phonon, we have changed the statement in the revised version. Here we emphasize that the behaviors of the LO phonon, the degeneracy with the TO phonon and linear dispersion, are integrated. Ref. [43] did not observe the linear dispersion of LO phonon near the CBZ due to the strong screening from Ni(111) substrate. Hence, it remains crucial to comprehensively observe both behaviors from the CBZ to the large momentum range.

Comment[#2-2]: *While I find the manuscript well-organized and easy to follow, there are still several important points that should be clarified:*

1. The polariton dispersion is modeled with the help of Eq. (1). The results of the fitting, where the authors find the dielectric response of the copper substrate $\epsilon_{\text{bot}}=4.45$, however, seem to be in conflict with the fact that copper should exhibit metallic behavior in the frequency region of interest. That questions either the validity of the model, or values of other parameters (S , r_{eff}), which are extracted for a free-standing monolayer and might not be valid for a monolayer on top of metal. The copper substrate might introduce strain, screening, and non-local effects at the atomistic level or yield the emergence of the so-called mirror/acoustic polaritons, where the polaritons in h-BN hybridize with their "mirror image" if there is a metal underneath. The role of the substrate might be rather non-trivial, so it could be crucial to perform the study with different substrates.

Reply[#2-2]: Thank you for your important comments. Indeed, for metals, it is unreasonable to expect a constant dielectric response. Therefore, we have removed the section describing substrate screening using the model.

To gain a better understanding of the screening effect, following the suggestion, we have added a comparative experiment with Cu single crystal substrates. We observed that, unlike the Cu foil, the linear dispersion of the LO phonon in monolayer h-BN is completely screened on the Cu single crystal substrate. The LO/TO phonon frequency at the CBZ on both substrates is 172.5 meV, ruling out the possibility of strain as the origin. We attribute the linear behavior of LO phonons in h-BN/Cu foil to the increased distance from h-BN caused by the surface roughness of the Cu foil substrate. The nonlocal screening strength weakens rapidly with increasing distance, thereby retaining

the linear behavior of the LO phonon.

In fact, our observations are consistent with the acoustic phonon polaritons predicted in [ACS Photonics 7, 2610-2617 (2020)], resulting from the hybridization of phonon polaritons in monolayer *h*-BN and their electromagnetic “mirror image” in the Cu foil substrate.

Changes

Page 5, line 136: “Screening effect from Cu foil” section and Fig. 3.

Page 7, line 208: “Computational analyses indicate that in the presence of a metal substrate, PhPs in a strict polar monolayer can undergo hybridization with their electromagnetic “mirror image,” resulting in the emergence of the so-called acoustic PhP^{33,40}. These acoustic PhPs exhibit larger confinement, stronger near-field enhancement, slower group velocities, and nearly identical polariton lifetimes when compared to conventional PhPs in freestanding monolayers³³. In our study, the observed dispersion and properties of the 2D PhPs align with the characteristics predicted for acoustic PhPs, thanks to the nonlocal screening provided by the Cu foil substrate. Nevertheless, to conclusively establish whether the observed PhPs are indeed acoustic in nature, more high-precision experiments are needed in the future.”

Comment[#2-3]: *2. Possibly related to the previous point: based on Fig. 3, the values of $\omega_{(TO/LO)}$ near $q=0$ found from the theoretical modeling do not match the experimental results very nicely.*

Reply[#2-3]: In our study, the LO/TO frequency of *h*-BN at the CBZ on both Cu foil and Cu single crystal substrate is ~ 172.5 meV, which is consistent with previous experimental reports [Nature Materials 20, 43-48 (2021)] [Nature 514, 209-212 (2014)]. Various studies have reported LO/TO phonon frequencies in the range of 161-172 meV at the CBZ, based on calculations using different parameters and methods [Physical Review B 80, 224301 (2009)] [Nature Materials 20, 43-48 (2021)] [Nano Letters 17, 3758-3763 (2017)].

It is important to note that the deviation between experimental and calculated values does not impact our analysis of the dispersion behavior of the LO phonon. *In the revised version, we have moved the DFT calculation results of Fig.3 to the Supplementary Information.*

Comment[#2-4]: *3. The authors emphasize some advancements they made to improve the HREELS setup compared to conventional arrangements. I am missing a schematic of their improvements. They should also mention the energy-momentum resolution achievable with their setup and comment on the data processing (e.g., extraction of the second-derivative spectra).*

Reply[#2-4]: We have included a new section titled “Comparison of 2D-HREELS and

conventional HREELS” along with Fig. S1 in the Supplementary Information, which provides a schematic comparing 2D-HREELS with conventional HREELS. We also added energy and momentum resolution and a brief description of data processing in the Methods. It is worth noting that, as we share the same analyzer as the mainstream angle-resolved photoemission spectroscopy (ARPES), our data processing follows the same procedures as in ARPES. Further information on the 2D-HREELS can be found in the original papers [Review of Scientific Instruments 86, 083902 (2015)].

Changes

SI: Add “Comparison of 2D-HREELS and conventional HREELS” section and Fig. S1.

Page 16, line 351: “A comparison of 2D-HREELS and conventional HREELS is shown in Fig. S1 in the Supplementary Information. With this setup, the ultimate energy and momentum resolutions are better than 0.7 meV and 0.002 Å⁻¹ (ref.²²). To compromise with the detection efficiency, we have set the resolutions to 3.3 meV and 0.025 Å⁻¹ for energy and momentum, respectively.”

Page 16, line 359: “2D-HREELS data were processed in the commercial software Igor Pro 9. The second-derivative spectra are represented in negative second derivative $-d^2I/d\omega^2 > 0$, where the I is the intensity and ω is the energy loss.”

Comment[#2-5]: *Other remarks:*

4. Introduction, lines 27 – 29: this sounds as over-claiming. This study only covers phonon dispersions in one representative 2D polar material, so I cannot see how this can be universal.

Reply[#2-5]: We have made modifications in the revised version to limit the scope of our study to *h*-BN.

Changes

Title: “Observation of the Exotic Behavior of Optical Phonons in Monolayer Hexagonal Boron Nitride”

*Page 2, line 30: “These exotic behaviors of the optical phonons in *h*-BN exhibit prospects in potential applications in future optoelectronics.”*

Comment[#2-6]: *5. 2D color plots are missing the color bars with the measured quantities, their values, and units.*

Reply[#2-6]: We have added the color bars in 2D color plots in Fig. 2, Fig. 3, and Fig. 4.

Comment[#2-7]: *6. Fig. 2 a,b show rather nonuniform intensities – this should be*

commented.

Reply[#2-7]: The nonuniform intensities originate from the surface roughness of the Cu foil substrate, the according explanations have been added to the revised version.

Changes

Page 4, line 87: “For the h-BN/Cu foil sample, it is important to note that the surface roughness of the Cu foil substrate introduces nonuniform background features in STM and LEED images. Due to the constraints associated with the commercial cold rolling process and the flexibility of thin Cu foil, the surface roughness of the substrate is unavoidable and serves as a tuning factor for the phonon properties of h-BN, as will be discussed later.”

Comment[#2-8]: *7. Fig. 2d: lines denoting phonon replicas are hardly visible.*

Reply[#2-8]: We have made the purple lines thicker.

Comment[#2-9]: *8. What is the meaning of vertical dashed lines in Fig. 3a?*

Reply[#2-9]: We have removed the dashed lines.

Comment[#2-10]: *9. As I understand, the gray area in Fig. 3b shows uncertainty in E-resolution. This should be explicitly mentioned as it also imposes uncertainties for the extracted points in Fig. 3b.*

Reply[#2-10]: The gray area does show the uncertainty in the energy resolution. In the revised version, we have explicitly mentioned it in the legend to Fig. 3.

Changes

*Page 12, line 286: “The height of the gray shaded areas in **b** and **e** represent the full width at half maximum of the zero-loss peak, reflecting the energy uncertainty of the measured data.”*

Comment[#2-11]: *10. Lines 143-144: there are studies with free-standing or encapsulated monolayers, so semi-infinite substrates are not the only option for applications.*

Reply[#2-11]: In the revised version, we have removed this sentence.

Comment[#2-12]: *11. Line 180: in literature, v_g typically denotes the group velocity, not its value normalized by c .*

Reply[#2-12]: We have corrected this error in the revised version.

Changes

Page 7, line 188: “ $v_g = \partial\omega/\partial q$ ”

Comment[#2-13]: *12. Line 221: SETM should be STEM.*

Reply[#2-13]: We have corrected this error in the revised version.

Changes

Page 8, line 233: “...but its flat spatial distribution enables s-SNOM/STEM-EELS to measure the dispersion of PhPs with high precision”.

Comment[#2-14]: *13. Fig. 4 a,b: I anticipate that the dark blue lines are extracted from the fit, while the dark blue dots are from the presented measurement – this should be explicitly mentioned.*

Reply[#2-14]: In the revised version, we have explicitly mentioned the meaning represented by the dark blue lines and dots.

Changes

Page 13, line 295: “Solid circles represent the experimental results of this work, empty squares represent the calculated results for a freestanding monolayer (ML), and empty rhombus represent the measured results of 10 nm-thick extracted from ref.³⁵. The solid curves represent the fitting results to the corresponding data.”

Comment[#2-15]: *14. The last section already discusses the comparison with other techniques for measuring PhPs. It would also be nice to explicitly state an effective q/d -range + q/d -resolution of the different techniques (HREELS, s-SNOM, STEM-EELS) in a table. Also, besides Refs. [19] and [33] dealing with PhPs in STEM-EELS, there is a more recent reference, which might be mentioned [Small 17, 2103404 (2021)].*

Reply[#2-15]: Thank you again for your detailed inspection and comments on our manuscript. We have added the recent reference [Small 17, 2103404 (2021)] as Ref. [44] to the revised version. Additionally, we have also added Table 1 to compare the measurement of PhPs using Raman spectroscopy, s-SNOM, STEM-EELS, and 2D-HREELS techniques.

Changes

Page 8, line 247: “A comparison of different experimental techniques for PhPs measurements is listed in Table 1.”

Page 14, line 303: “Table 1| Comparison of different experimental techniques for PhP

measurements.”

Reviewer #3 (Remarks to the Author):

Comment[#3-1]: *Review of "Observation of the Breakdown of Optical Phonon Splitting in a Two-dimensional Polar Monolayer" by Li et al.*

In this work, Li and coauthors present direct evidence that in a monolayer of hexagonal Boron Nitride, the longitudinal optical (LO) and transverse optical (TO) phonons at the zero center exhibit no energy splitting, contrary to the conventional LO-TO splitting seen in 3D materials. Despite extensive theoretical descriptions of this behavior, the authors claim this to be the first experimental verification of the LO-TO splitting breakdown. They then characterize the LO phonon (or, if you like, the phonon polariton) on a Copper foil substrate and describe the effect of screening on the momentum dependence near the zone center, along with its deceleration and confinement factors.

The work is interesting. To be honest I'm not sure how important the work is for the broader scientific community, as I will try to explain. The research appears to be thorough and claims are generally well-founded.

Reply[#3-1]: Thank you for your interest in our work. We have modified the statement on the observation of LO phonon behaviors of monolayer *h*-BN. Accordingly, we changed the title. In the revised version, we have highlighted the limitations of previous studies and emphasized the significance of observing the exotic behavior of LO phonons from the CBZ to a wide momentum range.

Changes

Page 1, line 1 (Title): "Observation of the Exotic Behavior of Optical Phonons in Monolayer Hexagonal Boron Nitride".

Page 2, line 47-67: We have changed the statements for Introduction.

Comment[#3-2]: *I provide more detailed comments below that may help the authors improve their manuscript further.*

*1) The authors claim that this is the first experimental observation of the absence of LO-TO splitting. They point to ref. 20 and say "a weak 2D PhP signal has been detected...the momentum window is too narrow to reveal the overall properties of the 2D PhPs." This reference appears to observe LO phonons in monolayer *h*BN having linear dispersions near the Brillouin zone center, consistent with this work. While I understand that the experimental measurement technique is different, why is it fair to claim that the result presented here is the first experimental observation of the absence of LO-TO splitting? It's quite clear from figure 1 and also from theory that the TO phonon should not be impacted by the dimensionality of the Coulomb interaction. Thus measuring the LO phonon frequency to be that near the TO frequency (at finite wavevector) all but guarantees that LO-TO splitting (at the Brillouin zone center) is absent. I wonder about how differentiable your results are from ref. 20 from the standpoint of verification of no LO-TO splitting.*

Reply[#3-2]: While linear dispersion was observed in ref.²⁰, it remains unverified whether LO and TO phonons are degenerate at the CBZ. This uncertainty arises from several factors: their measurements did not extend to the CBZ, they were unable to measure the TO phonon, and the quoted energy value for the TO phonon (~169.5 meV) was derived from the measurement of 3D bulk *h*-BN [Phys. Rev. 146, 543 (1966)], significantly lower than the experimental results for monolayer *h*-BN (~173 meV from [Nature Materials 20, 43-48 (2021)], ~172 meV from [Nat. Nanotechnol. 18, 529–534 (2023)], ~172.5 meV from our experiments). This discrepancy has also been noted in [Nature Materials 20, 43-48 (2021)]. In contrast, our experimental measurements cover the strict CBZ and extend continuously to a wide range of momenta. Therefore, our results explicitly demonstrate the experimental verification of theoretical predictions.

In the revised version, we have changed the statement to emphasize that our experiments comprehensively observe LO phonon behaviors in monolayer *h*-BN, including the breakdown of LO-TO splitting and the finite slope of LO phonons, from the CBZ to the large momentum range.

Comment[#3-3]: *2) Theoretical works cited in this manuscript have in some sense extensively studied the expected impact of a 2D Coulomb interaction on the LO-TO optical phonon dispersions. Since 2019, we have known that for polar 2D systems: 1. there should be no LO-TO splitting at the Gamma point, 2. the LO phonon should have a linear slope near the Gamma point and 3. the LO phonon is essentially a phonon polariton. All three of these conclusions fall out of the 2D nature of the Coulomb interaction in these monolayer or effectively aperiodic-in-z systems. Reading this paper, none of the results shown in figs. 1-3 were at all surprising.*

Reply[#3-3]: Since 2019, the behaviors of LO phonons in polar 2D systems are indeed well-understood theoretically. However, as we stated in the revised Introduction, experimental verification of the exotic behaviors of LO phonons from the CBZ to the large momentum range is still absent. Therefore, it remains crucial to conduct a comprehensive experimental investigation.

In the revised version, we have also added comparative experiments of Cu single crystal substrate in Fig. 3 to elucidate the unique role of the Cu foil substrate.

Comment[#3-4]: *My question is this. Why do the authors claim that this work is exceptionally useful? For example they state "Combined with first-principles calculations, it is demonstrated that this exotic behavior originates from the long-range Coulomb interactions of lattice vibrations." This statement is true, but it's hardly a result of this work. It has been known for years. They also state "Our experimental results demonstrate the correctness of the 2D implementation of the nonanalytical-term correction and lay the foundation for a correct understanding of the behavior of the LO phonons in 2D polar monolayers." Again, I find this misleading. The correct understanding of the behavior of the LO phonons in 2D polar monolayers has been*

known by the community for at least four years. One can argue that experimental verification was already provided in ref. 20 four years ago as well. Various follow up works involving phonon polaritons in 2D systems have been carried out since then. How can this paper then claim to lay the foundation of this field?

Reply[#3-4]: Thanks for pointing out this issue. In the revised version, we have changed the statements and highlighted the existing works.

Changes

Page 5, line 129: “The nonanalytic behavior of the LO phonon originates from the long-range Coulomb interaction caused by the polar lattice vibrations of monolayer h-BN. The modulation of phonon dispersions by long-range Coulomb interactions is greatly affected by dimensionality. Compared with the calculated LO phonon dispersion results of monolayer h-BN under traditional 3D boundary periodic conditions (Fig. S5 in SI), our measurement results give a comprehensive experimental verification of the 2D implementation method reported by Sohier et al.^{12,27,28}.”

Comment[#3-5]: *3) Following up on this, I don't find the theoretical calculations in this work particularly useful. In fact, they are in many ways directly reproducing those from ref. 12. Figure 3a in this manuscript is hardly different qualitatively from the comparison done in figure 2 in ref. 12. The authors use figure 3 to state that "This demonstrates that the long-range Coulomb interaction clearly affects the dispersion of the LO phonons in 2D polar materials." Why is this statement presented in such a way that it is a new finding? This is exactly the finding presented in ref. 12! I do not understand why the authors do not, at the very least, give proper credit for this finding to ref. 12, since this exact phenomena has been explained so extensively in their work. Instead, the current text makes it sound as if the authors were the first to discover this result.*

Reply[#3-5]: Thanks for point out this issue. In the revised version, we have modified the statements and highlighted the contribution of Sohier *et al.* In order to emphasize the experimental comparison of different substrates, we have removed the calculated curves from Fig. 3 of the main text. The new calculated results are presented in the Supplementary Information (“*Comparison of calculated optical phonon dispersions*” section and Fig. S5) to demonstrate the comparison between experiments and calculations.

Changes

Page 4, line 102 “To assign the phonon modes measured by 2D-HREELS, the calculated phonon dispersions, with the 2D implementation of the density-functional perturbation theory (DFPT) developed by Sohier et al.^{12,27,28}, are also superimposed in Fig. 2c and 2d (green curves, see Methods for details).”

Page 16, line 376 (Method): “To get the correct dispersion of the LO phonon, we performed the 2D implementation of DFPT developed by Sohier et al.^{12,27,28}.”

SI: Add “Comparison of calculated optical phonon dispersions” section and Fig. S5.

Comment[#3-6]: *4) In the fit presented in fig. 3b, the authors find that the effective substrate screening is 4.45. Can the authors contextualize what this means? The authors make it a point that the Cu foil used is a good substrate for modifying the polariton dispersion without breaking the induced electric field. Can the authors comment on what makes Cu foil good, and/or what material properties would be necessary to optimize substrate interactions for a specific dispersion or polariton property? Including this kind of analysis, and if possible any more experimental data with different substrates, to improve the manuscript's novelty quite a bit.*

Reply[#3-6]: Thank you for your valuable comments. In the revised version, we have added comparative experiments of Cu single crystal substrate. We observed distinctly different behaviors of LO phonons on the two substrates, indicating a significant alteration of the nonlocal screening strength. Considering the high sensitivity of nonlocal screening to the substrate-to-sample distance, we identified the surface roughness, the only difference between the two substrates, leading to the variation in screening strength. The surface roughness of the Cu foil results in an increased distance between the substrate and monolayer *h*-BN, consequently weakening the nonlocal screening strength and producing phonon polaritons with excellent characteristics.

Changes

We have accordingly made corresponding modifications in the “Screening effect from Cu foil” section and Fig. 3 of the main text.

Comment[#3-7]: *5) The authors conclusion includes idea that "monolayer h-BN/Cu foil is an ideal system for exploring light-matter interactions." I don't really understand what this means. "exploring light-matter interactions" is an extremely vague statement. What do the authors really mean? There are plenty of aspects about these polaritons that are never discussed, for example their expected lifetimes, which would greatly impact their potential for different light-matter-based applications. Can the authors please be more specific with their outlook.*

Reply[#3-7]: Thank you for your important comments. Unfortunately, our technique cannot provide information on the expected lifetime of phonon polaritons. But theory predicts [ACS Photonics 7, 2610-2617 (2020)] that nonlocal screening by metal substrates will have little effect on the expected lifetime of polaritons, but will result in stronger near-field enhancements. Based on the ultra-slow deceleration factor, ultra-high confinement factor and stronger near-field enhancement of the monolayer *h*-BN/Cu foil system, we have added specific examples in the Conclusion that are expected to facilitate the development of related applications.

Changes

Page 16, line 376: “This results in the 2D PhP of h-BN exhibiting an ultra-slow deceleration factor ($\sim 5 \times 10^{-6} c$) and ultra-high confinement factor (~ 4000) surpassing the 2D limit of freestanding h-BN. This advancement is expected to facilitate further development of optoelectronic applications such as subdiffraction imaging⁴⁸⁻⁵⁰, nanoresonators^{41,51}, single molecule detection⁴⁰, etc.”

Reviewer #4 (Remarks to the Author):

Comment[#4-1]: *This work provides experimental evidence, supported by DFPT calculations, of the breakdown of the TO-LO splitting in a monolayer of h-BN, used as a prototype of polar 2D materials. There are some strong points and some weak points that I discuss below.*

The strong points of this work are:

- Challenging 2D-HREELS measurements providing directly a 2D energy-momentum mapping of monolayers; this is the first experimental measurement of the 2D phonon dispersion of monolayers*
- Effect of the Cu foil substrate is new: finding of the ultra-slow group velocity and ultra-high confinement is unexpected*
- Methods, data analysis, calculations, interpretation and conclusions are sound*
- Very well written manuscript and high scholar presentation*
- Comparison of the methods for measuring phonon polaritons in the last paragraph is useful and appealing to a broad audience*

Reply[#4-1]: We would like to thank you for your positive review of our work.

Comment[#4-2]: *The weak points are:*

- the paper is entirely based on one measurement only (although very good), which is replotted in several different ways, but it is always the same measurement. The new data are only in figure 2c. Then figure 2d and 2e are extracted from that (which is normal and necessary), and figure 3a and 3b are zooms of the same data (again, this is normal and necessary), and figure 4 contains again the same data, with the comparison with literature (again, this is good). The SI do not show any new data, as well. From the discussion of the data I can imagine that with this technique the authors cannot study any dependence on the layer thickness nor on the type of substrate, but there are studies that should have been performed and could have been shown at least in the SI: the same measurement could have been performed on different materials or at least on different samples of the same material, even fabricated in the same way. The lack of any additional measurement poses a problem of reproducibility.

Reply[#4-2]: The main challenge in our method of using HREELS to study monolayer systems lies in the sample preparation, as it requires large-area single crystals of macroscopic size (~ 1 mm). To date, we have not been able to obtain large-area h-BN samples with multiple layers, preventing us from conducting thickness-dependent experiments. Thickness-dependent experiments are the direction of our future efforts, which may be obtained through sample transfer. In this manuscript, we focus on the study of monolayer materials and substrate screening.

Thank you for suggesting the idea of changing the substrate. We did prepare samples

of different substrates for comparison. Unfortunately, despite attempting a variety of substrates including SiO₂/Si(111), 4H-SiC(0001), and Cu(111) single crystal, only samples grown on Cu single crystals were successful. But the Cu single crystal substrate was also the most critical. In the revised version, we have added data on Cu single crystal substrates, which helps address comments about the nonlocal screening effects of substrates raised by the previous three reviewers.

In addition, we have conducted many experiments on samples of Cu foil and Cu single crystal substrates, all of which have exhibited good reproducibility. Data from the reproduction experiment are presented in Fig. S7 in the Supplementary Information.

Changes

We have accordingly made corresponding modifications in the “Screening effect from Cu foil” section and Fig. 3 of the main text.

Supplementary Information: We have added “Supplemental 2D-HREELS data of h-BN grown on Cu single crystal” section and Fig. S7 to demonstrate the good reproducibility of our measurements.

Comment[#4-3]: *- the noteworthy results are the demonstration of the breakdown of the TO-LO splitting in a monolayer of h-BN and the effect of the substrate; while the latter is new, it is the first one that is the most relevant for future studies, and unfortunately it is not entirely new, as it has already been demonstrated at gamma point. Indeed, the paper ‘Experimental demonstration of the suppression of optical phonon splitting in 2D materials by Raman spectroscopy’ published in 2D materials in 2020 has basically the same title of this paper (and the same introduction and motivation), but it is not discussed. That paper concerns Raman measurements of WSe₂ and h-BN (two different materials), and it also studies the thickness dependence and the dependence on the type of stacking. Although the work from Jiade Li et al. is more complete (and more difficult experimentally) because thanks to HREELS also measures phonons far from gamma, its aim is still to demonstrate the correctness of the theory in ref. 12, which has already been demonstrated at least at gamma by the other paper. This poses a problem of novelty.*

Reply[#4-3]: Thank you for mentioning the important reference. In the revised version, we have changed the statement of our manuscript and added a discussion of the reference as Ref.¹⁷.

We emphasize that “the breakdown of the TO-LO splitting in a monolayer” encompasses more than just the degeneracy of LO and TO phonons at the CBZ. Recent theoretical researches have clarified that it includes three distinct aspects: the degeneracy of LO and TO phonons, the finite slope of LO phonons, and the properties of phonon polaritons. It is important to note that previous verifications of LO phonon behaviors in polar monolayers were incomplete, and comprehensive verification from the CBZ to the large momentum range remains essential. In addition, our work also

investigated the nonlocal screening effects of the substrates, which enables *h*-BN to produce ultra-slow deceleration factor and ultra-high confinement factor beyond the 2D limit of freestanding *h*-BN, surpassing the highest record reported in the literature. Furthermore, we have introduced a new methodology for measuring PhPs in monolayer materials using 2D-HREELS. Given the ongoing trend of continuous synthesis of large-area 2D single crystal materials, the 2D-HREELS technique is expected to play an increasingly important role.

Changes

Page 3, line 56: “Raman scattering has revealed the degeneracy of LO and TO phonons at the CBZ¹⁷, but it fails to detect the linear dispersion of LO phonons due to the lack of momentum resolution.”

We have shifted our research focus from observing the breakdown of LO-TO splitting to observing the exotic behaviors of LO phonons:

Page 3, line 47-67: “In the last two decades, various theoretical models predict that the LO phonon degenerates with the TO phonon in 2D monolayers and exhibits a “V-shaped” nonanalytic behavior near the CBZ (Fig. 1c)¹⁰⁻¹⁴Hence, there is an urgent need for direct experimental investigation into the exotic behaviors of LO phonons and the properties of PhPs spanning from the CBZ to a large momentum range in strict 2D polar monolayers. Such research is crucial for both fundamental physics and potential applications.”

*Page 4, line 78: “Here, using monolayer *h*-BN as a prototypical example, and employing the state-of-the-art 2D-HREELS technique, we systematically observe the complete dispersion behaviors of the LO phonons from the CBZ to the Brillouin zone boundary, and investigate the properties of the 2D PhPs in polar monolayer systems.”*

*Page 8, line 251: “Through precise measurements of the phonon spectra in monolayer *h*-BN with high energy and momentum resolutions, our study systematically explores the behaviors of the LO phonon from the CBZ to a large momentum range.”*

Accordingly, to better summarize the study of our manuscript, we changed the title: “Observation of the Exotic Behavior of Optical Phonons in Monolayer Hexagonal Boron Nitride”

Comment[#4-4]: *- the DFPT calculations are useful and well performed, but they reproduce calculations that are very similar to those in ref.12, thus the main novelty of this paper remain the experimental finding (with the limitations just discussed)*

Reply[#4-4]: In the revised version, we have moved the calculation results from Fig. 3 to the Supplementary Information to better emphasize the experimental comparison of different substrates. In addition, to facilitate a more direct comparison between the experimental and calculation results, we modified the calculation content and added the calculation results of the full 3D periodic boundary conditions of different vacuum

distances. We also emphasize that the original work of Ref.¹² comes from Sohier *et al.*

Changes

SI: Add “Comparison of calculated optical phonon dispersions” section and Fig. S5.

Page 4, line 102 “To assign the phonon modes measured by 2D-HREELS, the calculated phonon dispersions, with the 2D implementation of the density-functional perturbation theory (DFPT) developed by Sohier et al.^{12,27,28}, are also superimposed in Fig. 2c and 2d (green curves, see Methods for details).”

Page 16, line 376 (Method): “To get the correct dispersion of the LO phonon, we performed the 2D implementation of DFPT developed by Sohier et al.^{12,27,28}.”

Comment[#4-5]: - *there is some ‘overselling’ in some parts of the manuscript. Examples:*

- the title refers to a 2D polar monolayer and suggests a universal finding, but the paper is only about h-BN. Although there is no reason why the results should not be applicable to other monolayers, I suggest writing h-BN in the title, otherwise the reader will think that more materials are measured*

- in the conclusions, ‘Combined with first-principles calculations, it is demonstrated that this exotic behavior originates from the long-range Coulomb interactions of lattice vibrations’ is not completely correct, because this has already been demonstrated both theoretically and experimentally. I would rather write ‘Our work provides additional experimental verification of the fact that this exotic behavior originated from, as already been predicted in ...’.*

Reply[#4-5]: In the revised version, we have added h-BN in the title and modified the statement in the Conclusion.

Changes

Title: “Observation of the Exotic Behavior of Optical Phonons in Monolayer Hexagonal Boron Nitride”

Page 8, line 251 (Conclusion): “Through precise measurements of the phonon spectra in monolayer h-BN with high energy and momentum resolutions, our study systematically explores the behaviors of the LO phonon from the CBZ to a large momentum range. The degeneracy of LO and TO phonons, as well as the “V-shaped” nonanalytic behavior of the LO phonon at the CBZ, are comprehensively demonstrated in our 2D spectra.”

Comment[#4-6]: - *Can the authors try to explain why the signal intensity of the TA mode is very weak? Is this a general finding in other materials studied with this technique? Is this expected?*

Reply[#4-6]: The selection rules of HREELS in the impact scattering regime determine the signal intensity. These rules dictate that when the lattice vibrations exhibit odd parity relative to the direction of measurement, the cross-section is zero. The TA mode is consistently odd parity relative to the strict Γ -M and Γ -K directions and therefore theoretically undetectable. This is widely recognized in the HREELS study [L. Vattuone, M. Rocca, L. Savio: High resolution electron energy loss spectroscopy (HREELS): a sensitive and versatile surface tool. In: Surface Science Techniques, Springer Ser. Surf. Sci., Vol. 51, ed. by G. Bracco, B. Holst (Springer, Berlin, Heidelberg 2013) pp. 499–529, Chap. 17].

However, in practical measurement, due to the finite width of the collection slit of the analyzer, signals in non-highly symmetry directions will inevitably be received, providing an opportunity to detect weak signals of the TA mode. The weak TA signals have been reported in graphene [Phys. Rev. Lett. 115, 075504 (2015)] [Phys. Rev. Lett. 131, 116602 (2023)].

The theory of HREELS cross-section (intensity) in the impact scattering regime is introduced in [Ibach H, Mills D. Electron energy loss spectroscopy and surface vibrations. Academic Press 1982].

Comment[#4-7]: - *line 120: remove 's' from exhibits*

Reply[#4-7]: Thanks. We have corrected the grammatical error there.

Comment[#4-8]: - *the discussion of the possible applications of these findings should be slightly expanded. Now it is quite vague and not appealing to a broad audience: which are the devices and the applications (and the fundamental studies) that would benefit from the strong light matter interaction?*

Reply[#4-8]: We have added in the Conclusion concrete examples where our research may lead to the development of applications.

Changes

Page 8, lin 256: "This results in the 2D PhP of h-BN exhibiting an ultra-slow deceleration factor ($\sim 5 \times 10^{-6} c$) and ultra-high confinement factor (~ 4000) surpassing the 2D limit of freestanding h-BN. This advancement is expected to facilitate further development of optoelectronic applications such as subdiffraction imaging⁴⁸⁻⁵⁰, nanoresonators^{41,51}, single molecule detection⁴⁰, etc."

Comment[#4-9]: - *In conclusion, I believe that the paper is very good and highly significant and as such worthy of being published in a very good journal. However, it lacks the necessary novelty, experimental reproducibility, and comparison with existing literature, necessary to be published in this high impact journal.*

Reply[#4-9]: Following the suggestions of all reviewers, we revised the manuscript as described above in detail. The highlight can be summarized as follows:

1. We have conducted comprehensive observations of the behaviors of LO phonon in monolayer *h*-BN, a representative 2D polar monolayer.
2. We have investigated the effect of nonlocal screening on the LO phonons of monolayer *h*-BN by comparing Cu foil and Cu single crystal substrates.
3. We have discovered record-breaking ultra-slow deceleration factor and ultra-high confinement factor for PhPs.
4. We have provided a new methodology for measuring PhPs in polar monolayers from CBZ to the large momentum range.

Reviewers' Comments:

Reviewer #1:

Remarks to the Author:

The authors have significantly revised their manuscript, notably providing additional measurements for monolayer hBN on a single crystal Cu substrate. This is a valuable addition in my opinion, as it sheds lights on the screening effects coming from Cu, which was my main concern in the first version of the manuscript. Indeed, the dispersion obtained on single crystal Cu is exactly what one would expect from a metallic substrate, with a vanishing slope of the LO dispersion. Comparing this with the results on Cu foil, the authors make a convincing argument that the foil's surface roughness is at the origin of the weakened screening observed initially. I believe the author's have satisfactorily adjusted their statements to reflect what can be fairly deduced from their experiments.

While the details of the screening mechanism warrants further exploration (from the authors and the community in general), the quality and value of the original experimental work remain. Thus, my concerns have been satisfactorily addressed, and I can recommend publication.

Reviewer #2:

Remarks to the Author:

Although many of the reactions to address the reviewers' comments are satisfactory, I still think the manuscript needs further revisions. I summarize my remaining concerns below:

1. I do not think the phonon dispersions observed in this work are particularly "exotic", and I would remove this phrase from the title and other parts of the paper.
2. Lines 118-119*: The new results clearly show that the difference in the substrate roughness impacts the phonon properties. However, I would not claim it is a "tuning" factor, as there is limited control over the roughness.
3. I highly appreciate the new dataset for a monolayer of hBN on a single-crystal Cu substrate. However, the difference between the results acquired for hBN on a rough foil and flat single crystal is only commented on. I am missing any theory, e.g. a qualitative model, that would confirm the validity of these explanations and the physics behind them: the model used to interpret the data acquired on the Cu foil was criticized by the reviewers, so it was simply removed from the manuscript. The dashed lines in Fig. 3 a,b,d,e are only guides to the eye, not a fit or a model – this is a major flaw for me.
4. As Fig. 3 is now split into two rows showing the results obtained for a foil and a crystal, I am missing the corresponding data for a Cu crystal in Fig. 2. A plot analogous to Fig. 2c for a crystal is hidden in the supplementary information as Fig. S6. I would also be interested in seeing a counterpart for Fig. 2d.
5. Somehow related to points 3 and 4: what does the bright spot in Fig. 3d for $q = 0$ and ~ 185 meV correspond to?
6. Lines 186 – 187*: the authors talk about "phonon slope", but it should be group velocity.
7. Lines 351 – 352*: nanoresonators are devices/platforms for applications, but they are not applications themselves; subdiffraction imaging is not an example of an optoelectronic application.
8. Panels (a) and (c) of Fig. S1 explaining the difference between 2D- and conventional HREELS are confusing. In particular, trajectories of the electrons through the hemispherical analyzer and their dispersion in energy should be sketched better so it is clear how the q -E measurement is obtained.

*Line numbers from the manuscript with tracked changes.

Reviewer #3:

Remarks to the Author:

Review #2 of "Observation of the Breakdown of Optical Phonon Splitting in a Two-dimensional Polar Monolayer" by Li et al.

The authors have done an adequate job of addressing my primary concerns. I appreciate their efforts and professionalism.

As a result, I find the updated manuscript much more satisfactory. I have included some additional comments that the authors may wish to consider. They are mostly intended to help improve the manuscript and its readability.

1. There were a number of instances where I felt that the grammar was overly distracting, I highlight some instances below. I understand there are numerous challenges with writing a manuscript, especially in a secondary language, so these are in no way meant to be mean, only constructive.

a) In a few instances, the authors use "degenerate" as a verb, apparently conveying the idea "become degenerate." Unfortunately this word in this meaning does not exist, I would suggest changing this to "become degenerate" or something similar, to avoid any confusion.

b) At the end of the abstract, I don't understand the idea of the last sentence. Particularly, the authors say "...phonons in h-BN exhibit prospects in potential applications in future optoelectronics." I would try to rephrase this.

c) I would suggest replacing "the momentum space" with "momentum space" in most instances.

d) On line 159, replace "is primarily originates" with "primarily originates"

2. In the first paragraph of the main text, the authors say "...the lattice symmetry guarantees that the LO and the transverse optical (TO) phonons are degenerate with zero slopes at the center..." I would suggest clarifying that the slopes referred to are those of the phonon dispersion.

3. I was a little puzzled as to why the authors, at the end of the introduction, say "we systematically observe the complete dispersion behaviors of the LO phonons from the CBZ to the Brillouin zone boundary..." In particular, would it be a stronger claim to say that using 2D-HREELS, you can study all unique phonon modes of hBN? Especially because you make a point that previous studies of 2D hBN phonons can't capture the TO phonon directly, and therefore cannot confirm the absence of LO/TO splitting. This appears as an important part of your study, as you differentiate from these alternative experimental techniques.

Reviewer #4:

Remarks to the Author:

Most of the concerns of reviewer 4 were properly addressed and, those were not addressed, cannot be addressed in a reasonable time.

Therefore, the paper can be accepted in its present form. It was greatly improved.

Point-by-point Response to Reviewers' Comments

We appreciate the time and effort of all four reviewers in reviewing our work. We are also grateful for the positive assessments of the revised version and the instructive suggestions provided by the reviewers. We have meticulously addressed each comment and implemented the necessary changes. With these revisions, we believe that all the comments have been addressed properly. The reviewers' comments are summarized in blue italic font. Our detailed point-by-point response are in regular black font. The revision in the manuscript are quoted in black italic font.

We submit two revised versions: the clean version, and the highlighted version with all the changes marked to aid the reviewers in further review.

Reviewer #1 (Remarks to the Author):

Comment[#1-1]: *The authors have significantly revised their manuscript, notably providing additional measurements for monolayer hBN on a single crystal Cu substrate. This is a valuable addition in my opinion, as it sheds lights on the screening effects coming from Cu, which was my main concern in the first version of the manuscript. Indeed, the dispersion obtained on single crystal Cu is exactly what one would expect from a metallic substrate, with a vanishing slope of the LO dispersion. Comparing this with the results on Cu foil, the authors make a convincing argument that the foil's surface roughness is at the origin of the weakened screening observed initially. I believe the author's have satisfactorily adjusted their statements to reflect what can be fairly deduced from their experiments.*

While the details of the screening mechanism warrants further exploration (from the authors and the community in general), the quality and value of the original experimental work remain. Thus, my concerns have been satisfactorily addressed, and I can recommend publication.

Reply[#1-1]: We thank you for the recommendation to publish our manuscript in Nature Communications. This will encourage us to further explore the details of the screening mechanism in the future.

Reviewer #2 (Remarks to the Author):

Comment[#2-1]: *Although many of the reactions to address the reviewers' comments are satisfactory, I still think the manuscript needs further revisions. I summarize my remaining concerns below:*

1. I do not think the phonon dispersions observed in this work are particularly "exotic", and I would remove this phrase from the title and other parts of the paper.

Reply[#2-1]: We have substituted the term "exotic" with "nonanalytic". We believe that this adjustment effectively distinguishes our observations from the conventional understanding of LO-TO splitting in 3D systems.

Changes

Title: "Observation of the Nonanalytic Behavior of Optical Phonons in Monolayer Hexagonal Boron Nitride"

Lines 25 – 27: "...we report the comprehensive and direct experimental verification of the nonanalytic behavior of LO phonons by inelastic electron scattering spectroscopy."

Lines 51 – 52: "On the other hand, it has been theoretically demonstrated that the nonanalytic behavior of LO phonons in 2D polar monolayers makes them equivalent to phonon polaritons (PhPs)¹⁵."

Lines 65 – 67: "Hence, there is an urgent need for direct experimental investigation into the nonanalytic behavior of LO phonons and the properties of PhPs spanning from the CBZ to a large momentum range in strict 2D polar monolayers."

Lines 123 – 125: "The dispersion of the LO phonon is unambiguously demonstrated and shows a distinct "V-shaped" nonanalytic behavior near the Γ point."

We retained the term "exotic" in the final sentence of the Abstract (Line 31) to emphasize the surprising nature of the ultra-slow group velocity and ultra-high confinement of phonon polaritons observed in our study.

Comment[#2-2]: *2. Lines 118-119*: The new results clearly show that the difference in the substrate roughness impacts the phonon properties. However, I would not claim it is a "tuning" factor, as there is limited control over the roughness.*

Reply[#2-2]: We have changed the statements in the revised version.

Changes

Lines 90 – 93: "Due to the constraints associated with the commercial cold rolling process and the flexibility of thin Cu foil, the surface roughness of the substrate is unavoidable. This aspect distinctly affects the phonon properties of h-BN, a point that will be further elucidated later in the subsequent discussions."

Comment[#2-3]: 3. I highly appreciate the new dataset for a monolayer of *h*BN on a single-crystal Cu substrate. However, the difference between the results acquired for *h*BN on a rough foil and flat single crystal is only commented on. I am missing any theory, e.g. a qualitative model, that would confirm the validity of these explanations and the physics behind them: the model used to interpret the data acquired on the Cu foil was criticized by the reviewers, so it was simply removed from the manuscript. The dashed lines in Fig. 3 a,b,d,e are only guides to the eye, not a fit or a model – this is a major flaw for me.

Reply[#2-3]: The screening effects of metal substrate have always been a matter of concern, but so far, there is no strict solution. Here we can discuss with a simple model. According to Sohler *et al.* [Nano Lett. 17, 3758–3763 (2017)], the dispersion of LO phonon of monolayer *h*-BN can be expressed as

$$\omega_{LO}(q) = \sqrt{\omega_{TO}^2(q=0) + \frac{S \cdot q}{\epsilon_{env} + r_{eff}q}}$$

with

$$\epsilon_{env} = \frac{\epsilon_{top} + \epsilon_{bot}}{2}$$

where ϵ_{env} is the average dielectric constant of the surrounding environment, and ϵ_{top} (ϵ_{bot}) is the effective dielectric constants of the top (bottom) side of the monolayer *h*-BN. The values and corresponding meanings of other parameters can be found in the Method section of the manuscript. In the case of our study, the top of monolayer *h*-BN is vacuum, we have $\epsilon_{top} = \epsilon_{vacuum} = 1$. The bottom of monolayer *h*-BN is Cu substrate. According to [Nano Lett. 20, 5, 2986–2992 (2020)], the dielectric function can be written in terms of image charges

$$\epsilon_{bot}(q, \omega) = \left(1 + \frac{1 - \epsilon_{Drude}(\omega)}{1 + \epsilon_{Drude}(\omega)} e^{-2qd}\right)^{-1}$$

with the Drude form of the dielectric function of the Cu substrate

$$\epsilon_{Drude}(\omega) = 1 - \frac{\omega_p^2}{\omega(\omega + i\gamma)}$$

where d is the distance between the monolayer *h*-BN and the Cu substrate. The values $\hbar\omega_p = 8880$ meV and $\hbar\gamma = 103$ meV of Cu are taken from [Nano Lett. 21, 2444–2452 (2021)].

Figure R1 shows the LO phonon dispersion curves of *h*-BN with different values of d , calculated using the model described above. It can be seen that as the distance d increases, the behavior of LO phonons changes from parabolic dispersion to linear dispersion. This is qualitatively consistent with the results of experimental

measurements.

Fig. R1. Calculated LO phonon dispersion curves of monolayer *h*-BN with different distance between *h*-BN and Cu substrate.

It should be noted that it is difficult to find an exact expression for the accurate nonlocal effects [Nano Lett. 20, 5, 2986–2992 (2020)] [Phys. Rep. 113, 195–287 (1984)] [Nat. Commun. 11, 366 (2020)]. Although the above simple model can qualitatively understand the experimental results, it does not give new insights into the nonlocal screening effects. Therefore, we believe that this model is not suitable for elaboration in the manuscript. As mentioned by reviewer #1, this open question awaits further efforts from the general community, based on our new experimental data.

Comment[#2-4]: *4. As Fig. 3 is now split into two rows showing the results obtained for a foil and a crystal, I am missing the corresponding data for a Cu crystal in Fig. 2. A plot analogous to Fig. 2c for a crystal is hidden in the supplementary information as Fig. S6. I would also be interested in seeing a counterpart for Fig. 2d.*

Reply[#2-4]: In the revised version, we have added the second derivative phonon spectra of the monolayer *h*-BN on Cu single crystal to Fig. S6 of the Supplementary Information, as a counterpart for Fig. 2d.

Comment[#2-5]: *5. Somehow related to points 3 and 4: what does the bright spot in Fig. 3d for $q = 0$ and ~ 185 meV correspond to?*

Reply[#2-5]: The bright spot is a result of the adsorption vibrations of carbon. Since the equipment we use to grow *h*-BN is also used to grow graphene, a small amount of carbon residue is unavoidable. However, as detailed in our manuscript (Lines 288 - 290) and demonstrated in Fig. S7 of the Supplementary Information, these slight adsorption signals do not affect the behavior of LO phonons of *h*-BN on Cu single crystals. The energy of *h*-BN LO phonons on both Cu foil and Cu single crystal at the Γ point remains

consistent, indicating that any differences in dispersion are solely attributed to the substrate's roughness, rather than possible stress or defects.

Comment[#2-6]: *6. Lines 186 – 187*: the authors talk about “phonon slope”, but it should be group velocity.*

Reply[#2-6]: Thank you for pointing this out. In the revised revision, we have replaced “slope” there with “group velocity”.

Changes

*Line 141: “Interestingly, our results indicate that the **group velocity** of the LO phonon...”*

Comment[#2-7]: *7. Lines 351 – 352*: nanoresonators are devices/platforms for applications, but they are not applications themselves; subdiffraction imaging is not an example of an optoelectronic application.*

Reply[#2-7]: We have changed the statements in the revised version.

Changes

Lines 259 – 260: “This advancement is expected to facilitate further development of subdiffraction imaging⁴⁸⁻⁵⁰, nanoresonators^{41,51}, and single molecule detection⁴⁰, among others⁵².”

Comment[#2-8]: *8. Panels (a) and (c) of Fig. S1 explaining the difference between 2D- and conventional HREELS are confusing. In particular, trajectories of the electrons through the hemispherical analyzer and their dispersion in energy should be sketched better so it is clear how the q-E measurement is obtained.*

**Line numbers from the manuscript with tracked changes.*

Reply[#2-8]: In Fig S1, we have revised the schematics of 2D-HREELS and conventional HREELS. The diagram illustrates the trajectory of electrons through the hemispherical analyzer and the process for obtaining a 2D energy-momentum spectrum. We are confident that these updated schematics enhance the comprehension of 2D-HREELS functionality. For more details and schematics about 2D-HREELS, please refer to references [Review of Scientific Instruments 86, 083902 (2015)] and [Physical Review Letters 131, 116602 (2023)].

Changes

Figure S1 (a) and (c) in Supplementary Information.

We sincerely appreciate your professional and thorough feedback, which has significantly contributed to the enhancement of our manuscript's quality.

Reviewer #3 (Remarks to the Author):

Comment[#3-1]: *Review #2 of "Observation of the Breakdown of Optical Phonon Splitting in a Two-dimensional Polar Monolayer" by Li et al.*

The authors have done an adequate job of addressing my primary concerns. I appreciate their efforts and professionalism.

As a result, I find the updated manuscript much more satisfactory. I have included some additional comments that the authors may wish to consider. They are mostly intended to help improve the manuscript and its readability.

Reply[#3-1]: Thank you for your positive feedback and constructive comments. We appreciate your recognition of our efforts. We have carefully considered and incorporated your additional comments below to further improve the manuscript.

Comment[#3-2]: *1. There were a number of instances where I felt that the grammar was overly distracting, I highlight some instances below. I understand there are numerous challenges with writing a manuscript, especially in a secondary language, so these are in no way meant to be mean, only constructive.*

a) In a few instances, the authors use “degenerate” as a verb, apparently conveying the idea “become degenerate.” Unfortunately this word in this meaning does not exist, I would suggest changing this to “become degenerate” or something similar, to avoid any confusion.

b) At the end of the abstract, I don’t understand the idea of the last sentence. Particularly, the authors say “...phonons in h-BN exhibit prospects in potential applications in future optoelectronics.” I would try to rephrase this.

c) I would suggest replacing “the momentum space” with “momentum space” in most instances.

d) On line 159, replace “is primarily originates” with “primarily originates”.

Reply[#3-2]: Thank you for your grammatical suggestions on our manuscript, we have made the following changes.

Changes

For a) We have avoided using ‘degenerate’ as a verb throughout the manuscript.

Lines 21 – 22: “Theoretical predictions propose that the LO phonon in 2D polar monolayers becomes degenerate with the TO phonon...”

Lines 48 – 50: “Over the last two decades, various theoretical models predict that, in 2D monolayers, the LO phonon is degenerate with the TO phonon, exhibiting a “V-shaped” nonanalytic behavior near the CBZ (Fig. 1c)¹⁰⁻¹⁴.”

Lines 125 – 126: “First, the LO and TO phonons are undoubtedly degenerate at the Γ

point (see also in Fig. 2e for the EDC at the Γ point)."

For b) We have rephrased this sentence.

Lines 31 – 33: "These exotic behaviors of the optical phonons in h-BN presents promising prospects for future optoelectronic applications."

For c) We have replaced "the momentum space" with "momentum space".

For d) We have replaced "is primarily originates" with "primarily originates".

Comment[#3-3]: *2. In the first paragraph of the main text, the authors say "...the lattice symmetry guarantees that the LO and the transverse optical (TO) phonons are degenerate with zero slopes at the center..." I would suggest clarifying that the slopes referred to are those of the phonon dispersion.*

Reply[#3-3]: We appreciate your suggestion for clarification. We have replaced "dispersion slopes" with "slopes" to make this point explicit and avoid any potential confusion.

Changes

Lines 39 – 41: "...the lattice symmetry guarantees that the LO and the transverse optical (TO) phonons are degenerate with zero dispersion slopes at the center of the Brillouin zone (CBZ) (Fig. 1a)."

Comment[#3-4]: *3. I was a little puzzled as to why the authors, at the end of the introduction, say "we systematically observe the complete dispersion behaviors of the LO phonons from the CBZ to the Brillouin zone boundary..." In particular, would it be a stronger claim to say that using 2D-HREELS, you can study all unique phonon modes of hBN? Especially because you make a point that previous studies of 2D hBN phonons can't capture the TO phonon directly, and therefore cannot confirm the absence of LO/TO splitting. This appears as an important part of your study, as you differentiate from these alternative experimental techniques.*

Reply[#3-4]: Thank you for your insightful comment. We agree that the strength of our study lies in the use of 2D-HREELS to investigate all unique phonon modes of monolayer h-BN, particularly in capturing the dispersion of both LO and TO phonons directly. This indeed distinguishes our work from previous studies.

In light of this comment, we have revised the statement in the Introduction to emphasize this point.

Changes

Lines 80 – 81: "...we systematically observe the complete dispersion behaviors of all unique phonon modes from the CBZ to the Brillouin zone boundary..."

Reviewer #4 (Remarks to the Author):

Comment[#4-1]: *Most of the concerns of reviewer 4 were properly addressed and, those were not addressed, cannot be addressed in a reasonable time.*

Therefore, the paper can be accepted in its present form. It was greatly improved.

Reply[#4-1]: We are glad to hear that you recommend our manuscript for acceptance. Thank you for your positive feedback on our revised version.

Reviewers' Comments:

Reviewer #2:

Remarks to the Author:

The authors' response and the additional changes in the manuscript have addressed all my remaining concerns. I recommend the manuscript for publication in Nature Communications.

Point-by-point Response to Reviewers' Comments

Reviewer #2 (Remarks to the Author):

Comment[#2-1]: *The authors' response and the additional changes in the manuscript have addressed all my remaining concerns. I recommend the manuscript for publication in Nature Communications.*

Reply[#2-1]: We thank you for the recommendation to publish our manuscript in Nature Communications.